# Evaluation of bread wheat (*Triticum aestivum* L.) genotypes for drought tolerance using morpho-physiological traits under drought-stressed and well-watered conditions

**Birhanu Mecha Sewore**[1,2]*, **Ayodeji Abe**[3]*, **Mandefro Nigussie**[4]

**1** Pan African University Life and Earth Science Institute (Including Health and Agriculture), University of Ibadan, Ibadan, Nigeria, **2** Department of Plant Sciences, College of Agricultural Sciences, Wachemo University, Hosanna, Ethiopia, **3** Department of Crop and Horticultural Sciences, Faculty of Agriculture, University of Ibadan, Ibadan, Nigeria, **4** Ethiopian Agricultural Transformation Institute, Addis Ababa, Ethiopia

* birhanu.mecha@yahoo.com (BMS); ayodabe@yahoo.com (AA)

**Data Availability Statement:** All relevant data are within the paper and its Supporting information files.

## Abstract

Increasing frequency of drought spells occasioned by changing climatic conditions, coupled with rise in demand for bread wheat, calls for the development of high yielding drought resilient genotypes to enhance bread wheat production in areas with moisture deficit. This study was designed to identify and select drought-tolerant bread wheat genotypes using morpho-physiological traits. One hundred and ninety-six bread wheat genotypes were evaluated in greenhouse and field experiments, under well-watered (80% of field capacity) and drought-stressed (35% of field capacity) conditions, for two years. Data were collected on five morphological traits (flag leaf size, flag leaf angle, flag leaf rolling, leaf waxiness and resistance to diseases) and 14 physiological traits. Relative water content (RWC), Excised leaf water retention (ELWR), Relative water loss (RWL), Leaf membrane stability index (LMSI), as well as Canopy temperature depression (CTD) at heading (CTDH), anthesis (CTDA), milking (CTDM), dough stage (CTDD) and ripening (CTDR) were estimated. Similarly, leaf chlorophyll content (SPAD reading) was recorded at heading (SPADH), anthesis (SPADA), milking (SPADM), dough stage (SPADD), and ripening (SPADR). Significant (p<0.01) genotypic differences were found for the traits under both well-watered and drought-stressed conditions. Associations of RWL with SPADH, SPADA, SPADM, SPADD and SPADR were significant (p<0.01) and negative under both watering regimes. The first three principal components accounted for 92.0% and 88.4% of the total variation under well-watered and drought-stressed conditions, respectively and comprised all the traits. The traits CTDD, CTDM, CTDR, SPADH, SPADA, SPADM, SPADD and SPADR with genotypes Alidoro, ET-13A2, Kingbird, Tsehay, ETBW 8816, ETBW 9027, ETBW9402, ETBW 8394 and ETBW 8725 were associated under both conditions. Genotypes with narrow flag leaves, erect flag leaf angles, fully rolled flag leaves, heavily waxed leaves, and resistant to disease manifested tolerance to drought stress. The identified traits and genotypes could be exploited in future breeding programmes for the development of bread wheat genotypes with tolerance to drought.

**Funding:** Author name: Birhanu Mecha Grant number: no African Union funded stipend, and field and lab research costs. The funders had no role in study design, data collection and analysis, decision to publish, or preparation of the manuscript.

**Competing interests:** The authors have declared that no competing interests exist.

# 1 Introduction

Bread wheat (*Triticum aestivum* L., 2n = 6x = 42, AABBDD) ranks first in total cultivated area and second in total production among cereal crops globally, is the third most important cereal food crop in the world after maize (Zea mays) and rice (Oryza sativa) [1]. It contains about 55% starch, 13% protein, as well as some dietary fats, vitamin B, zinc, calcium, and iron [2]. Globally, wheat is cultivated on about 218.22 million hectares, with an estimated total production of 765.53 million tonnes in the 2018/2019 cropping season [3]. Ethiopia is the largest wheat producer in sub-Saharan Africa [4], and currently, wheat is the second most important staple food crop in the country after tef [Eragrostistef. (Zucc.) Trotter] [5]. However, the demand for wheat in Ethiopia exceeds what the farmers produce. For example, in the 2020/2021 season, its production of 5.48 million tonnes fell short of the annual consumption of about 6.7 million tonnes [6]. In order to bridge the gap between its production and domestic demand, 1.503 million metric tonnes of wheat was imported into Ethiopia in 2021 [6]. The demand for wheat grain in Ethiopia has continued to rise due to the growing interest in its commercial and food value, especially in the production of local bread, porridge (genfo), local beer (tela), roasted grain (kolo), boiled grain (nifro), pasta, macaroni, and different confectionary products [7]. Despite its social and economic importance, wheat production in Ethiopia is constrained by adverse climatic conditions. Prominent among such constraints is the incidence of drought which results in reduced productivity and lowered grain yields [8].

In Ethiopia, more than 90% of wheat is grown under rain-fed conditions by small-holder farmers in the highly drought vulnerable arid and semi-arid farming areas. Drought stress leads to high yield losses [9]. Drought, which can occur at any growth and developmental stage [10], reduces plant growth and manifests in serious alterations to the crop's morphological, physiological, biochemical, and molecular processes [11, 12]. Depending on the crop's growth stage, intensity and duration of stress, and the nature of the crop's response mechanisms to stress, drought causes a decrease in relative water content, reduces leaf chlorophyll content, enhances stomatal closure, increases osmolytes and inhibits growth in wheat [13–15], which ultimately lead to reduced productivity. Gonfa and Tesfaye [16] had earlier reported that drought stress at tillering, flowering and grain filling resulted in 72%, 37% and 17% reduction in wheat grain yield, respectively. This suggests that selection for drought resilient wheat genotypes should be conducted at early stages of wheat growth and development [17]. However, under field conditions, multiple biotic and abiotic factors may coexist and confound results, leading to inconsistent conclusions. To better define drought stress tolerance, screening of genotypes under controlled conditions in the greenhouse could be used to complement screening on the field.

Nearly all crop species can tolerate water stress, but the levels of tolerance vary from species to species and even within species, because different species have different phenological, morphological, biochemical, physiological and molecular adaptation mechanisms [18]. Drought tolerance is not an easily quantifiable plant attribute. It is a complex physiological, morphological and molecular phenomenon. In wheat, drought tolerance is reported to be associated with relative water content, excised leaf water retention and relative water loss [19–21], leaf size, leaf angle, leaf rolling [19, 22], waxiness on leaf [23], canopy temperature depression, flag leaf chlorophyll content and leaf membrane stability index [19, 24]. Water deficit has been found to reduce the relative water content of plants [25]. Wheat yields are now anticipated to increase by 0.5 to 1.0% annually, which is less than the 2.0% needed to meet global demand [26]. Wheat production must at least double by the year 2030 in order to meet the world's expanding demand for the grain [27]. This suggests that between now and 2050, the average global wheat yield must increase from 2.93 t.ha-1 to 5.56 t.ha-1, with an annual growth rate of at least 1.6%

[28]. In contrast, Ethiopia must boost wheat production from its current 5.48 million metric tons to 10.50 million metric tonnes, with an average yield increase from 2.71 tons ha$^{-1}$ to 6.96 tons ha$^{-1}$ [6].The identification and selection of drought tolerant genotypes is an economically viable and biologically superior approach to boosting wheat production in areas with moisture deficit [29]. The testing of crop genotypes for drought tolerance based on their morpho-physiological response to drought stress may serve as the potent approach to the development of new cultivars [30]. However, since the reactions of bread wheat genotypes to moisture deficit are known to vary with growth stages, evaluating for morpho-physiological changes under drought stress could lead to genetic improvement of drought tolerant genotypes [31]. Information on morpho-physiological traits associated with enhanced wheat grain yields under moisture-stressed conditions is important for bread wheat improvement [32, 33]. Previous studies have revealed the possibility of improving wheat grain yield under drought stress by up to 50% using morpho-physiological traits as selection criteria [34]. For example, Geravandi et al. [35] reported excised leaf water retention (ELWR), relative water loss (RWL) and leaf membrane stability index (LMSI) as reliable indicators of drought tolerance in wheat. Also, the findings of [36, 37] and revealed that high relative water content (RWC) and low relative water loss (RWL) are important indicators of water status when screening wheat genotypes for drought stress tolerance. In another study [38], reported that leaf rolling is a potential proxy trait associated with water loss in cultivated wheat, and could be used for the for the screening of common wheat genotypes up to a certain level of moisture loss from the leaves.

The effects of drought stress on the morpho-physiological traits at different growth stages of cereal crops, including bread wheat, have been the subject of many studies [39]. However, there is paucity of information on morpho-physiological response of diverse Ethiopian bread wheat genotypes under drought-stressed conditions. Knowledge of such information could facilitate the development of drought tolerant cultivars for the moisture limited wheat growing areas of Ethiopia. This study was therefore conducted to screen a broad panel of bread wheat genotypes for drought tolerance using morpho-physiological characters. The specific objectives of the study were to: (i) evaluate 196 bread wheat genotypes for drought tolerance based on morpho-physiological traits, and (ii) assess the effectiveness, reliability and level of associations of the physiological traits in screening bread wheat genotypes for drought tolerance.

## 2 Methods

### 2.1 Description of study area

The research was conducted from 24th July to 31st of December during the 2020/21 and 2021/22 crop growing seasons at the Wachemo University Crop Research Farm Hosanna, Ethiopia (7˚33'13.9" N and 37˚53'2" E, 2177 masl). The genotypes were evaluated at field and greenhouse environments under two water regimes during 2020/2021 and 2021/2022 making four testing environments, from now referred to as Env1 (greenhouse 2020/2021), Env2 (field 2020/2021), Env3 (greenhouse 2021/2022), and Env4 (field 2021/2022). Hosanna has a mean annual precipitation of 721mm, and mean maximum and minimum temperatures of 26.04 ºC and 14.73 ºC, respectively. The soil of experimental field was clay loam with pH 6.81. The day/ night greenhouse's temperatures were 30ºC/ 20ºC, while the humidity ranged from 50 to 60%. The average monthly weather data for the periods of the field trials at the experimental site is presented in the S1 Table.

### 2.2 Genetic materials

The genetic materials comprised of 188 bread wheat genotypes and eight standard checks sourced from the Ethiopian Institute of Agricultural Research. The genotypes were selected

based on their yield performance under drought prone areas at national yield trials. The details of the eight standard checks are presented in S2 Table, while lists of the 188 bread wheat genotypes are presented in S3 Table.

## 2.3 Experimental treatments, design and crop establishment

The 196 bread wheat genotypes were evaluated under drought-stressed and well-watered condition in the greenhouse and field using a $14 \times 14$ lattice design with two replicates. On the field, experimental plots consisted of six rows which were 2.5 m long with between row spacing of 0.2 m. Plots were separated by 0.5 m. Two seeds were sown per hole spaced 0.1 m apart within the rows and later thinned to one plant per hole three weeks after emergence to give a plant population density of 500,000 plants/ha. For data collection, the border two rows were left and the middle four rows were used. All plants were watered regularly to maintain the soil moisture at 80 to 100% of field capacity (FC) from sowing to 75% heading. Thereafter, the drought-stressed trial was subjected to terminal drought stress by watering at 35% FC, while the well-watered trial continued to receive water at 80–100% FC till physiological maturity. In the greenhouse experiments, plastic pots which were 25 cm in diameter and 40 cm deep were used as experimental units. Each pot was filled with 5.0 kg soil. Ten seeds were sown per pot and later thinned to five plants per pot at three weeks after emergence. The experiment was replicated two times to make a total of 392 pots per water regime. The soil moisture content was monitored using a soil moisture probe (tensiometer) inserted in the middle of the pot and plot for greenhouse and field trials, respectively. On the field, fertilisers in the form of NPS (N: P2O5: S) 19-38-7 and urea were applied at the rate of 125 kg ha-1 and 150 kg ha-1, respectively. The full dose of NPS was applied at sowing, while urea was applied in three equal splits of 50 kg ha-1 at sowing, 30 and 50 days after sowing (DAS). In the greenhouse, 2.0 g each of NPS and urea were applied to each pot at sowing and an additional 1.0 g of urea per pot was applied at 50 DAS. Planting was done 24thJuly 2020 and 21st July 2021. Weeds were controlled manually by regular hand weeding. Propiconazole 250 g/L EC was used to control fungal diseases such as rust, following manufacturer's recommendations.

## 2.4 Data collection/trait measurements

All the morpho-physiological traits observation were recorded from ten randomly selected and tagged representative plants per plot, for field experiments and from five plants per pot, for the greenhouse experiment. The average per plot for the field, and per pot for the greenhouse experiments were used for statistical analyses.

**2.4.1 Measurement of leaf morphological traits.** *2.4.1.1 Flag leaf size.* The size of the flag leaf of each genotype was observed, scored and classified into small = 3, intermediate = 5, large = 7 and very large = 9 (https://www.genesys-pgr.org/descriptorlists/0bf081e6-522c-4261-8531-0d40638b2eb3).

*2.4.1.2 Flag leaf angle.* The angle of flag leaves of each genotype at anthesis was assessed visually, scored and classified using the visual scale of [40], where: 1 = droopy,2 = semi-droopy, 3 = semi-erect and 4 = erect.

*2.4.1.3 Flag leaf rolling.* The flag leaf rolling habit of each genotype was visually observed, scored and classified using the scale of [40], where: 1 = no rolling or outward leaf rolling, 2 = weakly-rolled, 3 = semi-rolled or twisted type leaf rolling and 4 = fully-rolled or inward leaf rolling.

*2.4.1.4 Waxiness of leaf.* The waxiness of the leaves of each genotype was observed visually, scored and classified into waxy = 1, intermediate = 5, not waxy = 9 and mixture = 10 leaved (https://www.genesys-pgr.org/descriptorlists/0bf081e6-522c-4261-8531-0d40638b2eb3).

*2.4.1.5 Disease severity score.* Disease severity was scored using the scale described by [41], where: 0 = highly resistant (HR), 1 = resistant (R), 3 = moderately resistance (MR), 5 = moderately susceptible (MS), 7 = susceptible (S) and 9 = highly susceptible (HS).

**2.4.2 Data collection of physiological traits.** *2.4.2.1 Canopy temperature depression (CTD).* The CTD, a measure of the difference between the ambient temperature (Ta) and canopy temperature (Tc), was for each plot in the field and pot in the greenhouse using a hand-held portable infra-red thermometer (IRT) at about 50 cm above the canopy and at an angle nearly 45˚ to the horizontal. Positive CTD values indicated canopies were cooler than the air. The data for each plot were the means of four readings per plot. The measurements were acquired on cloudless days between 10:00 am and 2:00 pm at heading, anthesis, milking, dough and ripening stages under bright sunlight by focusing lenses at the centre of a leaf canopy in each plot. To minimize the confounding effects of changes in environmental conditions on genotypic performance, images of all the genotypes within a replication were acquired within a 20 minute period. Mathematically, CTD was calculated by subtracting the average canopy temperature from the mean ambient temperature using the formula:

$$CTD \, (^{\circ}C) = Ta\text{–}Tc$$

Where: CTD = canopy temperature depression, Ta = ambient temperature, Tc = canopy temperature

*2.4.2.2 Flag leaf chlorophyll content (SPAD 502 plus meter readings).* Flag leaf chlorophyll content was measured using a chlorophyll meter (SPAD-502 plus meter, Konica Minolta Sensing, Inc. Japan). The SPAD readings were recorded on the flag leaves of ten plants per plot on the field and five plants per pot in the greenhouse at heading, anthesis, milking, dough and ripening. Three measurements at random locations in the middle of the flag leaf were made for each plant, and the average sample was used for analysis.

*2.4.2.3 Relative water content in percent (RWC %).* The RWC (%) of flag leaves was determined 10 days after anthesis (DAA). Flag leaves of ten plants from each plot on the field and five plants from each pot in the greenhouse were collected in sealed bags and immediately placed on ice and quickly transferred to the laboratory. The fresh weights (FW) of the samples were taken, and the leaves thereafter placed in test tubes containing distilled water maintained at 4˚C for 24 hours. After soaking, leaves were quickly and carefully blotted on tissue paper and the turgid weight (TW) measured. The samples were thereafter oven-dried at 80˚C for 24 hours to obtain their dry weight (DW). The fresh, turgid and dry weights of the flag leaves were used to calculate the RWC according to the following formula used by [42].

$$RWC(\%) = \, ((FW - DW))/((TW - DW)) \times 100.$$

Where: FW = Fresh weight of leaves, TW = Turgid weight of leaves, DW = Dry weight of leaves

*2.4.2.4 Excised leaf water retention (ELWR).* Flag leaves of ten plants per plot on the field and five plants per pot in the greenhouse were collected 10 DAA and the FW taken. The leaves were thereafter soaked in distilled water in test tubes for four hours, then wilted at 25˚C and reweighed to obtain the wilted weight of leaves after four hours (WW4h). The ELWR was computed using the formula [43]:

$$ELWR(\%) = [1 - ((FW - WW4h)/FW)] \times 100,$$

Where: FW = Fresh weight of leaves, WW4h = wilted weight of leaves after 4hrs at 25˚C.

*2.4.2.5 Relative water loss (RWL).* Flag leaves of ten plants per plot on the field and five plants per pot in the greenhouse were collected at 10 DAA and the FW taken. The leaves were

thereafter wilted for 4 hours at 25˚C, reweighed to obtain the WW4h, oven dried for 24 h at 80˚C and the DW measured. Leaf RWL was calculated according to the formula used by [44, 45]:

$$RWL(\%) = [((FW - WW4h))/(FW - DW)] \times 100.$$

Where: FW = Fresh weight of leaves, WW4h = wilted weight of leaves after 4hrs at 25˚C, DW = Dry weight of leaves.

*2.4.2.6 Leaf membrane stability index (MSI).* The LMSI was determined by measuring the electrical conductivity of leaf leachates in double distilled water at 40˚C and 100˚C [46]. Leaf strips (0.2 g) were cut into discs of uniform sizes and taken into test tubes containing 10 mL of double distilled water in two sets. The test tubes in one set was kept at 40˚C for 30 minutes and electrical conductivity (EC) of the water containing the sample was measured (C1) using a conductivity bridge. The test tubes in the other set were incubated at 100˚C in a water bath for 15 minutes and the EC measured (C2) using a conductivity meter. The LMSI was calculated using the formula given by [47] as follows:

$$LMSI(\%) = [1 - (C1/C2)] \times 100$$

Where:

LMSI = Leaf membrane stability index
C1 = electrical conductivity of the water containing the sample in set one.
C2 = electrical conductivity of the water containing the sample in set two.

## 2.5 Data analyses

All data were subjected to statistical analyses. The morphological data were analysed separately for the drought-stressed and well-watered treatments computed using [48]. Data on physiological traits were subjected to analysis of variance (ANOVA) for a lattice design using PROC GLM in SAS [49]. Analysis was conducted for each water regime and thereafter combined for the water regimes. Pearson's correlation coefficients were estimated to determine the relationships between pairs of traits using PROC CORR in SAS [49]. Principal Component Analysis (PCA) was carried out separately for the drought-stressed and well-watered conditions to obtain information on traits most effective in discriminating among the wheat genotypes. Principal Components (PCs) with Eigen values greater than or equal to one (≥1) were retained in the analysis. To further explain the relationships between the PCs and traits, a PCA biplot analysis of the first two PCs was conducted. The PCA analysis was done in R (CRAN R Software package version 0.97, R Core Team, 2022).

## 3. Results

### 3.1. Morphological traits

**3.1.1 Flag leaf size.** Three categories of flag leaf sizes: large (56 genotypes), intermediate (91 genotypes) and small (49 genotypes) were identified among the genotypes under drought-stressed condition (Fig 1A). Under drought-stressed condition, no genotype had very large flag leaf size (Fig 1A). However, under well-watered condition, the four categories of leaf sizes were found: very large (22 genotypes), large (69 genotypes), intermediate (78 genotypes) and small (27 genotypes) (Fig 2A).

**3.1.2 Flag leaf angle.** Three categories of flag leaf angles were found among the genotypes under both drought-stressed and well-watered conditions. No genotype had semi-droopy flag leaf angle. Under drought-stressed condition, 51 (26.3%), 97 (49.2%), and 48 (24.5%)

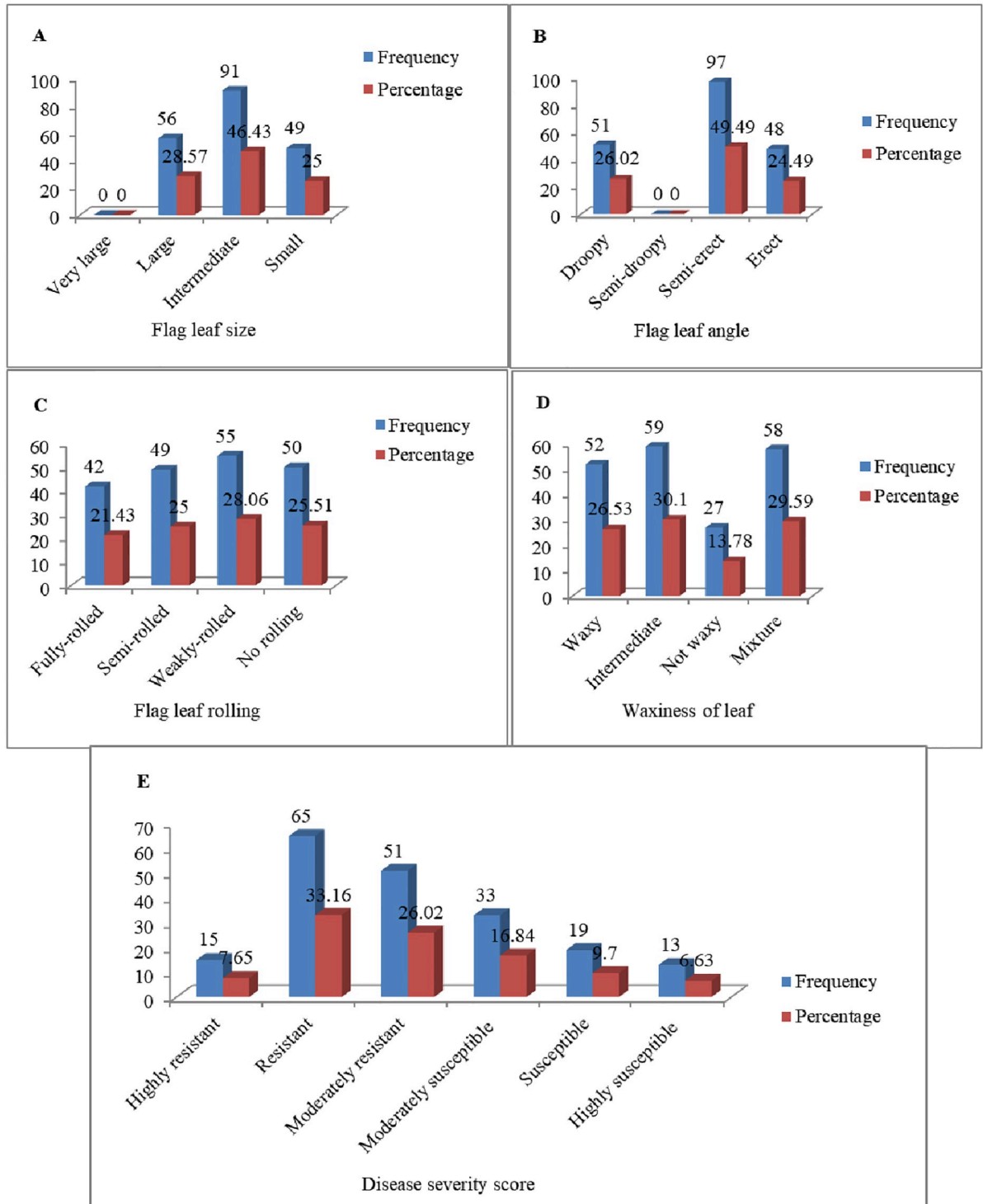

**Fig 1. Graphical illustration of morphological traits under drought-stressed conditions.**

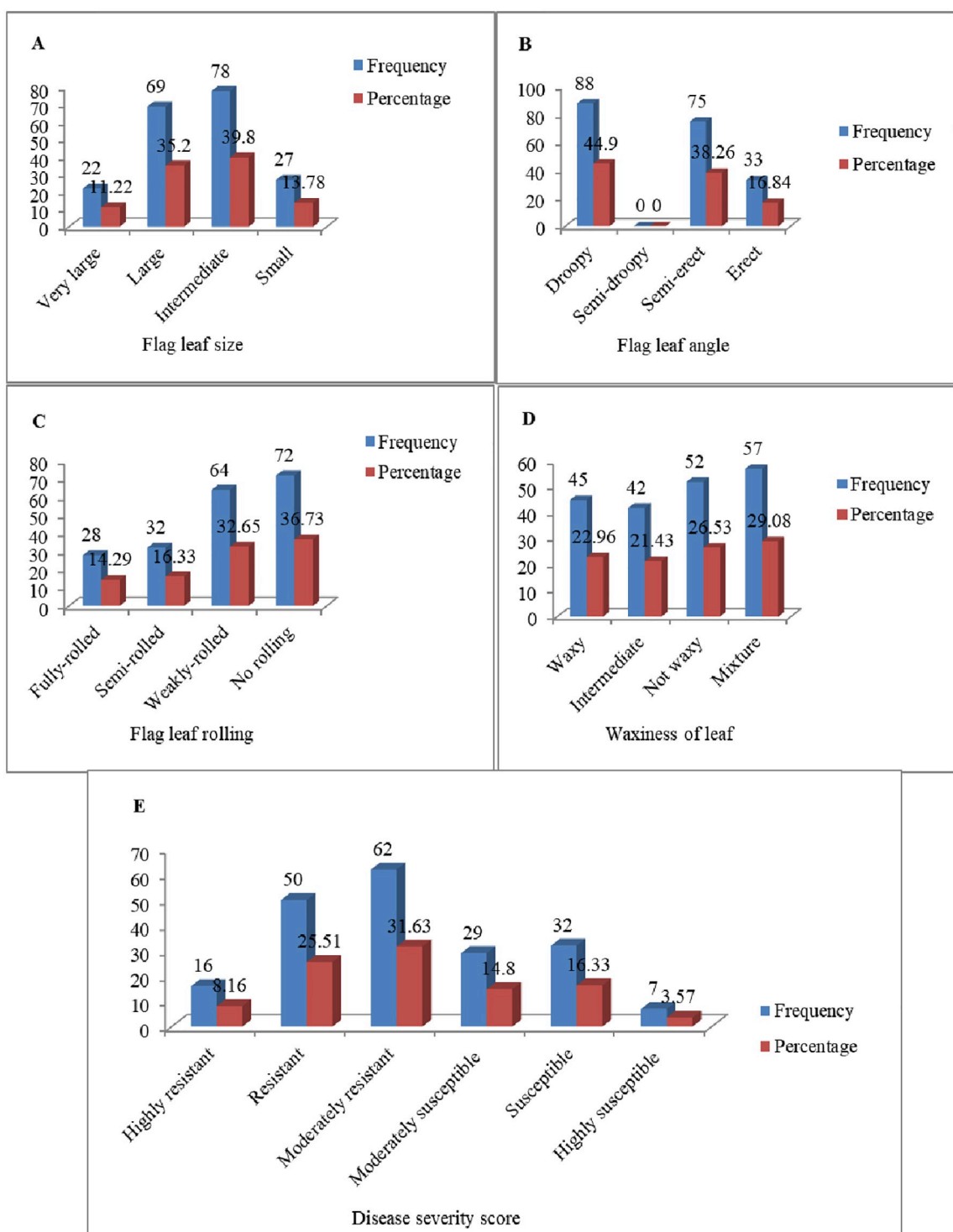

**Fig 2. Graphical illustration of morphological traits under well-watered conditions.**

genotypes had droopy, semi-erect and erect flag leaf angles, respectively (Fig 1B). However, under well-watered conditions, the flag leaf angles of 88 genotypes (44.9%) were droopy, 75 (38.3%) semi-erect, and 33 (16.8%) erect (Fig 2B). The result of this study revealed that flag leaf angle drooping occurs due to two main reasons: water stress and over watering. In this study, leaf drooping genotypes number increased under well-watered conditions (88 geno-types) relative to drought-stressed conditions (51 genotypes), indicating there was over water-ing. In this study, bread wheat genotypes with large flag leaves had droopy and weakly-rolled to no-rolled leaves than narrow-leaved genotypes; however, the genotypes with small flag leaves had erect flag leaves under both well-watered and drought-stressed conditions.

**3.1.3 Flag leaf rolling.** Out of the 196 wheat genotypes, 50 (25.5%) manifested no flag leaf rolling, 55 (28.1%) were scored weak rolling, 49 (25.0%) exhibited semi-rolling, while 42 (21.4%) had full rolling flag leaves under drought-stressed conditions (Fig 1C). Distribution for flag leaf rolling among the genotypes under well-watered condition showed that 72 (36.7%) of genotypes did exhibit no flag leaf rolling, 64 (32.7%) of genotypes had weak rolling, 32 (16.3%) of genotypes had semi-rolling, whereas 28 (14.3%) genotypes manifested full rolling (Fig 2C).

**3.1.4 Waxiness of leaf.** Genotypes that fell into the four categories of leaf waxiness were found. Twenty seven genotypes representing 13.8% of germplasm were not waxy, 59 (30.1%) had intermediate wax, 52 (26.5%) were classified to be strong waxy, and 58 representing 29.6% were classified as mixture under drought-stressed condition (Fig 1D). Under well-watered condition, 52 genotypes representing 26.5% had no wax, 42 (21.4%) genotypes were interme-diate, 45 (23.0%) genotypes had strong wax, and 57 genotypes representing 29.1% classified as mixture (Fig 2D).

**3.1.5 Disease severity score.** Based on the disease severity score, the 196 bread wheat genotypes were classified in to six categories: 15 (7.7%) of genotypes were highly resistant, 65 genotypes (33.2%) were resistant, 51 genotypes (26.0%) were moderately resistant, 33 (16.8%) genotypes showed moderate susceptibility, 19 (9.7%) of the genotypes were susceptible, and 13 (6.6%) of the genotypes were highly susceptible under drought-stressed conditions (Fig 1E), while 16 (8.16%) of genotypes were highly resistant, 50 (25.51%) of genotypes resistant, 62 (31.63%) of genotypes were moderately resistant, 29 (4.80%) of genotypes showed moderate susceptibility, 32 (16.33%) of genotypes were susceptible, and 7 (3.57%) of genotypes showed high susceptibility under well-watered conditions (Fig 2E).

## 3.2 Physiological traits

**3.2.1 Analysis of variance under drought-stressed and well-watered conditions.** Signifi-cant differences were found among genotypes under both drought-stressed and well-watered conditions for the studied traits canopy temperature depression (CTD) at heading, anthesis (CTDA), milking (CTDM), dough (CTDD), ripening (CTDR), SPAD at heading (SPADH), anthesis (SPADA), at milking (SPADM), dough (SPADD), ripening (SPADR), relative water content (RWC), excised leaf water retention (ELWR) and leaf membrane stability index (LMSI) (Table 1). Similarly, the effects of environment under both drought-stressed and well-watered conditions were significant for all the traits, except LMSI under drought-stressed con-dition. Genotype × environment interaction effects were significant for all the traits under both moisture regimes, except EWLR and LMSI under drought-stressed condition. The signif-icant genotype × environment interaction effect indicated that the genotypes responded differ-ently over the environments. Results of the mean square of the year and replication × years had highly significant variations for all the traits under both drought-stressed and well-watered conditions (Table 1). There were also highly significant differences observed for the mean

**Table 1. Mean squares from analysis of variance of 14 physiological traits of 196 bread wheat genotypes evaluated under drought-stressed and well-watered conditions in two years.**

| Sources of variation | DF | CTDH | CTDA | CTDM | CTDD | CTDR | SPADH | SPADA | SPADM | SPADD | SPADR | RWC | ELWR | RWL | LMSI |
|---|---|---|---|---|---|---|---|---|---|---|---|---|---|---|---|
| | | | | | | well-watered conditions | | | | | | | | | |
| Year | 1 | 3.1** | 4.6** | 3.7** | 2.9** | 2.6** | 384.5** | 438.0** | 367.7** | 325.2** | 293.6** | 863.8** | 881.6** | 561.5** | 670.9** |
| Rep(Years) | 2 | 3.6** | 6.1** | 4.0** | 3.3** | 2.8** | 42.4** | 78.6** | 41.7** | 49.2** | 34.9** | 308.5** | 322.7** | 457.4** | 346.6** |
| Genotype | 195 | 1.5** | 1.7** | 0.9 | 0.7** | 0.4** | 5** | 55.0** | 47.9** | 47.1** | 40.0** | 160.7** | 100.9** | 29.2$^{ns}$ | 341** |
| Env | 3 | 5.64** | 7.82** | 4.62** | 3.44** | 2.37** | 1512** | 1890 | 1461** | 1289** | 1076** | 23703** | 24046** | 2974** | 40453** |
| Gen*Env | 585 | 0.411* | 0.5* | 0.292* | 0.22* | 0.13* | 9.3** | 10.7** | 9.77** | 9.41** | 8.99** | 46.6* | 33.85** | 53.60** | 87.00** |
| Error | 785 | 0.162 | 0.24 | 0.17 | 0.12 | 0.065 | 6.5 | 8.2 | 7.18 | 6.92 | 6.35 | 38.24 | 31.67 | 34.8 | 65.72 |
| CV | | 10.3 | 9.1 | 9.7 | 10.2 | 9.6 | 6.2 | 5.8 | 5.3 | 6.7 | 7.1 | 9.5 | 9.0 | 10.8 | 8.5 |
| H$^2$BS | | 74.08 | 81.45 | 65.53 | 62.31 | 68.40 | 74.69 | 85.75 | 82.46 | 82.54 | 76.03 | 48.95 | 48.91 | 47.28 | 68.40 |
| | | | | | | drought-stressed conditions | | | | | | | | | |
| Year | 1 | 3.5** | 5.0** | 2.8** | 2.4** | 1.8** | 237.1** | 284.4** | 178.5** | 215.0** | 189.2** | 70520** | 69948** | 7550** | 1127** |
| Rep(Years) | 2 | 17.3** | 26.8** | 22.3** | 20.5** | 16.1** | 295.9** | 361.4** | 215.7** | 197.4** | 182.6** | 448.2** | 242.5** | 100.9** | 560.3** |
| Genotype | 195 | 1.42** | 1.92** | 1.29** | 0.78** | 0.59** | 66.0** | 68.55** | 63.0** | 62.25** | 60.43** | 143.21** | 95.1** | 44.50** | 518.25** |
| Env | 3 | 5.0** | 5.61** | 4.69** | 4.28** | 4.15** | 782.4** | 839.0** | 538.0** | 492.5** | 437.8** | 227.2** | 693.0** | 184.93** | 0.38$^{ns}$ |
| Gen*Env | 585 | 0.40** | 0.563** | 0.32** | 0.233** | 0.155** | 13.51** | 15.54** | 12.81** | 12.46** | 12.10** | 38.75** | 15.34ns | 39.00** | 23.97$^{ns}$ |
| Error | 785 | 0.11 | 0.185 | 0.122 | 0.094 | 0.078 | 6.56 | 7.78 | 6.71 | 6.52 | 6.04 | 14.93 | 29.92 | 27.8 | 107.87 |
| CV | | 9.6 | 8.7 | 8.8 | 8.3 | 7.9 | 6.2 | 8.5 | 8.1 | 7.0 | 8.4 | 8.7 | 9.4 | 11.7 | 10.3 |
| H$^2$BS | | 80.23 | 83.62 | 75.90 | 76.05 | 75.34 | 82.19 | 86.24 | 85.02 | 78.98 | 79.52 | 77.71 | 69.12 | 38.02 | 87.29 |

*;** = Significant at p < 0.05 and p<0.01probabilitylevels, respectively,

ns = Not significant, DF = degree of freedom; CTDH = canopy temperature depression at heading, CTDA = canopy temperature depression at anthesis, CTDM = canopy temperature depression at milking, CTDD = canopy temperature depression at dough, CTDR = canopy temperature depression at ripening, SPADH = SPAD heading, SPADA = SPAD anthesis, SPADM = SPAD milking, SPADD = SPAD dough, SPADR = SPAD ripening, RWC = Relative water content, ELWR = excised leaf water retention, RWL = Relative water loss, LMSI = leaf membrane stability index, CV = coefficient of variation, H$^2$BS = Heritability in broad sense.

square of the year and replication × years from combined analysis of data (Table 2). In the combined analysis of variance, effects of genotype, water regime and environment were significant for all the 14 physiological traits considered (Table 2). Non-significant genotype × water regime interaction effects were found for all the traits, indication that the response of the genotypes did not differ with water regimes. Similarly, the interaction effects of genotype × environment were non-significant for all the traits, except RWL. The three-way genotype × water regime × environment interaction effects were non-significant for all the physiological traits investigated.

Heritability in broad sense (H2BS) values ranged from 47.28% for RWL to 85.75% for the trait SPADA under well-watered conditions while a range of 43.02% for trait RWL to 87.29% for the trait LMSI was observed under drought-stressed conditions. Across the water regimes, the values of heritability in broad sense ranged between 43.02% and 87.29%. CTDA, SPADA and SPADM had high heritability estimates (above 79%) under both drought-stressed and well-watered conditions. RWL had the lowest broad sense heritability (<45%) under both drought-stressed and well-watered conditions.

### 3.3 Physiological response of the genotypes

The maximum and minimum performed genotypes for each trait ranked under both well-watered and drought-stressed conditions were presented in Table 3. The mean of physiological

**Table 2. Mean squares from combined analysis of variance of 14 physiological traits of 196 bread wheat genotypes evaluated under drought-stressed and well-watered conditions in two years in Ethiopia.**

| Sources of variation | DF | CTDH | CTDA | CTDM | CTDD | CTDR | SPADH | SPADA | SPADM | SPADD | SPADR | RWC | ELWR | RWL | LMSI |
|---|---|---|---|---|---|---|---|---|---|---|---|---|---|---|---|
| Years | 1 | 3.7** | 39.5** | 2.2** | 4.3** | 1.4** | 302.4** | 592.5** | 147.6** | 216.3** | 442.4** | 70890** | 69743** | 7854** | 5798** |
| Rep(Years) | 2 | 16.8** | 27.0** | 22.8** | 22.1** | 16.5** | 231.3** | 318.5** | 112.6** | 98.9** | 334.8** | 478.1** | 230.6** | 104.2** | 144.7** |
| Gen | 195 | 2.36** | 2.55** | 1.38** | 0.97** | 0.75** | 107.8** | 107.8** | 99.80** | 97.30** | 84.11** | 246.44** | 156.00** | 90.52** | 679.13** |
| WR | 1 | 2.34** | 3.43** | 3.10** | 1.42** | 1.22** | 860** | 2179** | 47.91** | 182.64** | 294.9** | 132.85** | 271.04** | 36.49** | 50328** |
| Env. | 3 | 2.47** | 15.91** | 5.09** | 6.58** | 4.54** | 818.3** | 659*** | 345.16** | 931.03*** | 782.6** | 32150** | 24695** | 2611** | 20330** |
| Gen x WR | 195 | 0.196ns | 0.304ns | 0.225ns | 0.15ns | 0.007ns | 6.77ns | 7.62ns | 6.58ns | 7.01ns | 7.49ns | 24.01ns | 19.69ns | 10.15ns | 58.4ns |
| Gen x Env | 585 | 0.040ns | 0.053ns | 0.037ns | 0.045ns | 0.041ns | 6.19ns | 6.60ns | 5.42ns | 5.68ns | 6.58ns | 17.41ns | 16.90ns | 62.54** | 13.06ns |
| WR x Env | 3 | 1.92** | 2.08** | 1.96** | 1.44** | 1.09** | 702.2** | 737.5** | 1847.4** | 2412** | 227.2** | 455.6** | 837.1** | 6599** | 23934** |
| Gen x WR x Env | 585 | 0.20ns | 0.324ns | 0.1533ns | 0.198ns | 0.082ns | 8.11ns | 8.87ns | 8.15ns | 8.29ns | 8.66ns | 20.17ns | 21.09ns | 25.43ns | 31.06ns |
| Error | 1568 | 0.3160 | 0.3934 | 0.1564 | 0.2188 | 0.1047 | 10.2374 | 10.8849 | 10.1538 | 10.0047 | 9.619 | 36.5225 | 28.5388 | 33.4975 | 73.1889 |
| CV | | 17.57 | 14.89 | 15.20 | 16.07 | 17.58 | 7.38 | 7.06 | 6.65 | 8.49 | 8.99 | 8.20 | 8.99 | 17.08 | 14.00 |

*,** = Significant at $p < 0.05$ and $p < 0.01$ probability levels, respectively;

ns = Not significant, DF = degree of freedom; CTDH = canopy temperature depression at heading, CTDA = canopy temperature depression at anthesis, CTDM = canopy temperature depression at milking, CTDD = canopy temperature depression at dough, CTDR = canopy temperature depression at ripening, SPADH = SPAD heading, SPADA = SPAD anthesis, SPADM = SPAD milking, SPADD = SPAD dough, SPADR = SPAD ripening, RWC = Relative water content, ELWR = excised leaf water retention, RWL = Relative water loss, LMSI = leaf membrane stability index, CV = coefficient of variation.

**Table 3. Estimates of maximum, minimum and mean of checks for the 14 physiological traits.**

| Traits | WR | Genotype | Maximum | Genotype | Minimum | Mean of checks | Grand mean | Reduction (%) due to DSC) |
|---|---|---|---|---|---|---|---|---|
| | | | | | | | | Mean |
| CTDH | WW | ETBW8491 | 4.32 | ETBW8984 | 2.28 | 2.94 | 2.83 | 21.5 |
| | DS | ETBW8491 | 3.05 | ETBW8984 | 1.52 | 2.46 | 2.24 | |
| CTDA | WW | ETBW9088 | 5.06 | ETBW8984 | 2.83 | 4.21 | 4.11 | 19.8 |
| | DS | ETBW8491 | 4.2 | ETBW8984 | 2.2 | 3.61 | 3.3 | |
| CTDM | WW | Alidoro | 4.12 | ETBW9441 | 2.13 | 3.47 | 3.19 | 17.4 |
| | DS | Alidoro | 3.68 | ETBW9449 | 1.95 | 2.97 | 2.81 | |
| CTDD | WW | ETBW8725 | 3.11 | ETBW8984 | 1.81 | 2.63 | 2.51 | 19.3 |
| | DS | ETBW9088 | 2.92 | ETBW9438 | 1.56 | 2.53 | 2.28 | |
| CTDR | WW | ETBW8725 | 2.42 | ETBW8981 | 1.41 | 1.99 | 1.89 | 12.6 |
| | DS | ETBW9088 | 2.35 | Menze | 1.15 | 1.85 | 1.72 | |
| SPADH | WW | Alidoro | 58.53 | ETBW9441 | 42.62 | 52.33 | 50.14 | 21.4 |
| | DS | Alidoro | 52.47 | ETBW9441 | 38.03 | 44.87 | 43.35 | |
| SPADA | WW | Alidoro | 60 | ETBW9441 | 43.91 | 53.94 | 51.89 | 16.5 |
| | DS | Alidoro | 53.59 | ETBW9441 | 39.12 | 45.26 | 44.43 | |
| SPADM | WW | Alidoro | 55.03 | ETBW9441 | 39.95 | 49.71 | 47.79 | 18.1 |
| | DS | Tsehay | 50.39 | ETBW9441 | 36.82 | 43.4 | 42.17 | |
| SPADD | WW | Alidoro | 51.02 | ETBW9441 | 36.31 | 45.6 | 44.02 | 18.7 |
| | DS | Alidoro | 47.88 | ETBW9449 | 33.9 | 40.77 | 39.58 | |
| SPADR | WW | Alidoro | 44.94 | ETBW9441 | 31.36 | 39.64 | 38.36 | 15.9 |
| | DS | Alidoro | 41.9 | ETBW8881 | 29.23 | 36.2 | 34.46 | |
| RWC | WW | Alidoro | 76.75 | ETBW8945 | 56.78 | 68.61 | 65.48 | 15.3 |
| | DS | Alidoro | 67.6 | Katar | 51.57 | 66.89 | 63.12 | |
| ELWR | WW | ETBW9088 | 65.4 | ETBW8945 | 49.46 | 58.56 | 55.5 | -14.3 |
| | DS | ETBW172872 | 63.3 | Digelu | 43.56 | 57.11 | 53.35 | |
| RWL | WW | ETBW8983 | 44.83 | ETBW172872 | 31.7 | 36.44 | 38.55 | 23.8 |
| | DS | ETBW8261 | 40.55 | Alidoro | 30.1 | 38.77 | 39.96 | |
| LMSI | WW | ETBW9088 | 77.21 | ETBW8827 | 55.00 | 70.06 | 65.99 | 20.2 |
| | DS | Alidoro | 69.13 | ETBW9441 | 37.75 | 57.71 | 51.9 | |

DSC = Drought-stressed condition, Max. = Maximum, Min. = Minimum, CTDH = canopy temperature depression at heading, CTDA = canopy temperature depression at anthesis, CTDM = canopy temperature depression at milking, CTDD = canopy temperature depression at dough, CTDR = canopy temperature depression at ripening, DS = drought-stressed condition, WW = Well-watered condition, SPADH = SPAD heading, SPADA = SPAD anthesis, SPADM = SPAD milking, SPADD = SPAD dough, SPADR = SPAD ripening, RWC = Relative water content, ELWR = excised leaf water retention, RWL = Relative water loss, LMSI = leaf membrane stability index, and CV = coefficients of variation.

traits recorded for the top 15 best performing and the bottom five performing genotypes out of 196 genotypes in terms of relative water loss (RWL) under drought-stressed condition are presented in Tables 4 and 5 under the both water regimes. Under well-watered condition, the genotypes differed significantly for all the physiological traits, except RWL. The canopy temperature depression (CTD) at heading (CTDH) of the genotypes under well-watered condition ranged from 2.28 (ETBW8984) to 4.32°C (ETBW8491) with a mean of 2.83°C (Table 3). The CTDH of 41.7% of the genotypes was higher than the mean of the standard checks (2.98°C). The CTDH of genotype ETBW8984 (2.28°C) was not significantly different from the values

**Table 4. Mean of physiological traits ranked according to their lower RWL performance under drought-stressed conditions for the top 15 best performing genotypes out of the 196 genotypes evaluated under both water regimes across the test environments.**

| Traits | WR | Top fifteen genotypes | | | | | | | | | | | | | | | Grand Mean |
|---|---|---|---|---|---|---|---|---|---|---|---|---|---|---|---|---|---|
| | | Alidoro | BW172938 | BW8491 | Bolo | BW172872 | Dinknesh | BW9088 | BW8870 | BW172936 | ET13A2 | Abola | BW8492 | BW8303 | BW8725 | BW9027 | |
| CTDH | NWT | 3.04 | 3.04 | 4.32 | 3.19 | 2.91 | 2.83 | 3.78 | 3.27 | 3.45 | 3.26 | 3.45 | 3.47 | 3.17 | 3.57 | 3.18 | 2.83 |
| | DST | 2.73 | 2.68 | 3.05 | 2.54 | 2.68 | 2.23 | 2.90 | 2.84 | 2.77 | 2.52 | 2.47 | 2.82 | 2.64 | 2.83 | 2.70 | 2.24 |
| CTDA | NWT | 4.60 | 4.00 | 4.97 | 4.72 | 4.20 | 4.06 | 5.06 | 4.32 | 4.55 | 4.81 | 4.57 | 4.26 | 4.74 | 4.67 | 4.23 | 4.11 |
| | DST | 4.12 | 3.85 | 4.20 | 3.61 | 3.98 | 3.35 | 4.02 | 3.91 | 3.94 | 6.95 | 3.59 | 3.59 | 4.07 | 3.97 | 3.85 | 3.30 |
| CTDM | NWT | 4.32 | 3.99 | 3.63 | 3.21 | 3.84 | 3.52 | 3.98 | 3.54 | 3.86 | 3.93 | 3.84 | 3.48 | 3.76 | 3.77 | 3.76 | 3.19 |
| | DST | 3.68 | 3.33 | 3.12 | 2.78 | 3.40 | 3.01 | 3.39 | 3.19 | 3.34 | 3.47 | 3.37 | 2.78 | 3.05 | 3.49 | 3.37 | 2.81 |
| CTDD | NWT | 2.74 | 2.63 | 3.05 | 2.86 | 2.73 | 2.72 | 2.92 | 2.88 | 3.02 | 2.96 | 2.70 | 2.61 | 2.72 | 3.11 | 2.84 | 2.51 |
| | DST | 2.67 | 2.72 | 2.83 | 2.44 | 2.80 | 2.36 | 2.92 | 2.82 | 2.91 | 2.77 | 2.54 | 2.69 | 2.67 | 2.85 | 2.91 | 2.28 |
| CTDR | NWT | 2.21 | 2.08 | 2.42 | 2.04 | 2.14 | 2.12 | 2.40 | 2.32 | 2.29 | 2.20 | 2.21 | 2.08 | 2.26 | 2.42 | 2.32 | 1.89 |
| | DST | 2.18 | 2.07 | 2.09 | 1.71 | 2.22 | 1.82 | 2.35 | 2.29 | 2.20 | 2.18 | 2.00 | 1.90 | 2.10 | 2.18 | 2.34 | 1.72 |
| SPADH | NWT | 58.53 | 53.34 | 55.59 | 53.47 | 55.33 | 55.42 | 56.06 | 55.18 | 50.77 | 54.07 | 54.74 | 55.06 | 53.11 | 54.96 | 54.52 | 50.14 |
| | DST | 52.47 | 48.21 | 46.00 | 42.65 | 49.33 | 46.23 | 47.95 | 46.82 | 47.17 | 49.76 | 48.11 | 42.41 | 45.78 | 49.04 | 48.06 | 43.35 |
| SPADA | NWT | 60.00 | 57.52 | 55.91 | 52.97 | 52.60 | 55.74 | 57.41 | 55.22 | 57.34 | 56.90 | 56.55 | 54.73 | 56.53 | 56.84 | 56.20 | 51.89 |
| | DST | 53.59 | 48.76 | 47.02 | 43.80 | 50.44 | 47.12 | 48.71 | 47.74 | 42.67 | 50.08 | 49.69 | 43.23 | 46.64 | 50.61 | 49.04 | 44.43 |
| SPADM | NWT | 55.03 | 51.09 | 52.59 | 50.74 | 52.79 | 52.33 | 52.96 | 52.60 | 48.54 | 51.31 | 51.79 | 52.45 | 50.61 | 52.80 | 51.80 | 47.79 |
| | DST | 47.80 | 46.24 | 44.99 | 41.72 | 47.97 | 44.21 | 46.63 | 45.30 | 46.29 | 47.99 | 46.94 | 40.94 | 44.53 | 47.80 | 46.90 | 42.17 |
| SPADD | NWT | 51.02 | 48.76 | 47.63 | 45.31 | 48.63 | 47.91 | 48.93 | 47.66 | 48.76 | 48.16 | 48.53 | 46.99 | 48.35 | 48.61 | 48.15 | 44.02 |
| | DST | 47.88 | 43.57 | 42.17 | 38.96 | 44.93 | 41.29 | 44.45 | 43.65 | 42.68 | 45.39 | 44.27 | 38.86 | 41.56 | 45.40 | 44.24 | 39.58 |
| SPADR | NWT | 44.94 | 42.13 | 41.77 | 40.17 | 43.38 | 41.73 | 42.59 | 41.83 | 44.48 | 42.17 | 43.13 | 41.05 | 43.44 | 42.87 | 42.57 | 38.36 |
| | DST | 41.90 | 38.08 | 36.26 | 33.96 | 40.31 | 36.16 | 40.37 | 38.32 | 38.52 | 40.10 | 40.03 | 33.85 | 37.85 | 40.31 | 39.30 | 34.46 |
| RWC | NWT | 76.75 | 71.33 | 73.82 | 70.86 | 73.54 | 73.11 | 72.73 | 72.73 | 70.75 | 74.56 | 67.41 | 67.35 | 70.23 | 70.16 | 69.95 | 65.48 |
| | DST | 67.62 | 65.25 | 62.17 | 60.17 | 60.24 | 65.17 | 59.69 | 63.58 | 63.73 | 61.17 | 61.54 | 59.43 | 58.71 | 60.93 | 64.42 | 63.12 |
| ELWR | NWT | 60.39 | 60.54 | 62.51 | 58.21 | 63.27 | 58.89 | 63.37 | 60.81 | 60.61 | 61.75 | 57.53 | 57.45 | 58.13 | 58.77 | 59.27 | 55.5 |
| | DST | 55.08 | 54.82 | 53.14 | 49.71 | 55.73 | 48.59 | 54.66 | 55.62 | 53.05 | 50.58 | 51.32 | 51.17 | 53.28 | 53.38 | 51.36 | 53.35 |
| RWL | NWT | 33.40 | 35.69 | 35.27 | 35.15 | 31.7 | 35.87 | 32.50 | 34.72 | 32.72 | 40.05 | 33.94 | 37.40 | 32.80 | 36.31 | 40.76 | 38.55 |
| | DST | 30.13 | 35.39 | 34.29 | 33.54 | 34.52 | 32.92 | 34.01 | 32.91 | 33.27 | 32.42 | 31.92 | 35.44 | 32.58 | 34.14 | 33.68 | 39.96 |
| LMSI | NWT | 69.39 | 74.98 | 76.90 | 73.78 | 75.73 | 74.38 | 77.21 | 76.01 | 74.76 | 61.10 | 72.26 | 71.35 | 75.57 | 74.35 | 62.54 | 65.99 |
| | DST | 69.13 | 64.78 | 66.38 | 61.96 | 65.28 | 60.17 | 65.07 | 65.31 | 62.38 | 61.04 | 60.07 | 60.95 | 64.08 | 63.23 | 61.55 | 51.9 |

CTDH = canopy temperature depression at heading, CTDA = canopy temperature depression at anthesis, CTDM = canopy temperature depression at milking,

CTDD = canopy temperature depression at dough, CTDR = canopy temperature depression at ripening, SPADH = SPAD heading, SPADA = SPAD anthesis, SPADM = SPAD milking,

SPADD = SPAD dough, SPADR = SPAD ripening, RWC = Relative water content, ELWR = excised leaf water retention, RWL = Relative water loss, LMSI = leaf membrane stability index, and

CV = coefficients of variation.

**Table 5. Mean of 14 physiological traits of the bottom five least performing genotypes out of the 196 genotypes evaluated under both water regimes across the test environments, ranked according to their higher relative RWL performance under drought-stressed condition.**

| TRAITS | WR | Bottom five genotypes | | | | | Grand Mean | Minimum | Maximum |
|---|---|---|---|---|---|---|---|---|---|
| | | Doddota | ETBW8984 | ETBW9441 | K6290Bulk | Galema | | | |
| CTDH | NWT | 2.46 | 2.28 | 2.29 | 2.60 | 2.43 | 2.83 | 2.28 | 4.32 |
| | DST | 1.83 | 1.52 | 1.54 | 2.10 | 1.78 | 2.24 | 1.52 | 3.05 |
| CTDA | NWT | 3.86 | 2.83 | 3.64 | 4.19 | 3.95 | 4.11 | 2.86 | 5.06 |
| | DST | 2.91 | 2.20 | 2.53 | 3.33 | 2.92 | 3.30 | 2.20 | 4.20 |
| CTDM | NWT | 2.72 | 2.76 | 2.13 | 3.11 | 2.74 | 3.19 | 2.13 | 4.12 |
| | DST | 2.56 | 2.37 | 1.99 | 2.65 | 2.28 | 2.81 | 1.95 | 3.68 |
| CTDD | NWT | 2.12 | 1.81 | 2.25 | 2.42 | 2.35 | 2.51 | 1.81 | 3.11 |
| | DST | 1.96 | 1.72 | 1.59 | 2.17 | 2.05 | 2.28 | 1.56 | 2.92 |
| CTDR | NWT | 1.49 | 1.42 | 1.59 | 1.81 | 1.70 | 1.89 | 1.41 | 2.42 |
| | DST | 1.35 | 1.15 | 1.26 | 1.53 | 1.39 | 1.72 | 1.15 | 2.35 |
| SPADH | NWT | 45.53 | 46.08 | 42.62 | 49.51 | 45.69 | 50.14 | 42.62 | 58.53 |
| | DST | 40.93 | 39.29 | 38.03 | 42.88 | 39.29 | 43.35 | 38.03 | 52.47 |
| SPADA | NWT | 46.94 | 47.88 | 43.91 | 51.14 | 47.38 | 51.89 | 43.91 | 60.00 |
| | DST | 42.07 | 40.66 | 39.12 | 43.60 | 40.01 | 44.43 | 39.12 | 53.59 |
| SPADM | NWT | 43.76 | 44.13 | 39.95 | 47.12 | 43.67 | 47.79 | 39.95 | 55.03 |
| | DST | 40.19 | 39.06 | 36.82 | 41.10 | 37.57 | 42.17 | 36.82 | 50.39 |
| SPADD | NWT | 40.03 | 40.38 | 36.31 | 43.67 | 40.80 | 44.02 | 36.31 | 51.02 |
| | DST | 37.69 | 36.15 | 34.75 | 38.17 | 35.42 | 39.58 | 33.90 | 47.88 |
| SPADR | NWT | 34.67 | 33.75 | 31.36 | 37.98 | 35.72 | 38.36 | 31.36 | 44.94 |
| | DST | 32.00 | 30.00 | 30.01 | 33.07 | 29.68 | 3.46 | 29.23 | 41.90 |
| RWC | NWT | 63.86 | 61.65 | 62.73 | 61.62 | 60.37 | 65.48 | 56.78 | 76.75 |
| | DST | 53.18 | 52.81 | 50.42 | 50.72 | 52.48 | 63.12 | 45.33 | 67.60 |
| ELWR | NWT | 54.94 | 51.67 | 53.33 | 52.32 | 51.82 | 55.5 | 49.46 | 65.37 |
| | DST | 46.52 | 45.16 | 44.26 | 44.26 | 46.17 | 53.35 | 38.82 | 55.73 |
| RWL | NWT | 35.22 | 43.03 | 35.31 | 36.76 | 37.85 | 38.55 | 31.70 | 44.83 |
| | DST | 39.79 | 40.57 | 41.96 | 38.64 | 41.13 | 39.96 | 30.13 | 42.25 |
| LMSI | NWT | 72.08 | 55.63 | 73.56 | 69.07 | 67.03 | 65.99 | 55.00 | 77.21 |
| | DST | 42.10 | 38.19 | 37.75 | 45.86 | 37.85 | 51.9 | 37.75 | 69.13 |

CTDH = canopy temperature depression at heading, CTDA = canopy temperature depression at anthesis, CTDM = canopy temperature depression at milking, CTDD = canopy temperature depression at dough, CTDR = canopy temperature depression at ripening, SPADH = SPAD heading, SPADA = SPAD anthesis, SPADM = SPAD milking, SPADD = SPAD dough, SPADR = SPAD ripening, RWC = Relative water content, ELWR = excised leaf water retention, RWL = Relative water loss, LMSI = leaf membrane stability index, and CV = coefficient of variation.

for genotypes ETBW9441 (2.29°C) and ETBW8797 (2.30°C) under well-watered condition (Table 3 and S5 Table). Under drought-stress conditions, the CTDH of the genotype ETBW8491 was the highest (3.05°C), while the CTDH of the genotype ETBW8984 showed the lowest (1.52°C) as presented in Table 3 and S4 Table. About 28.1% of the genotypes had a CTDH value greater than the mean for the standard checks (2.46°C) under drought-stressed conditions. The mean percentage reduction in CTDH due to drought stress ranged from 29.4% for genotype ETBW8491 to 33.3% for genotype ETBW8984. In the present study, across all genotypes, drought stress reduced CTDH by 21.22%. The CTDH response of the genotypes

to drought stress was most pronounced in ETBW8984 and ETBW9441 and least in ETBW8491 and Tsehay (S4 Table).

The mean CTD at anthesis (CTDA) under drought-stressed and well-watered conditions were 3.30˚C and 4.11˚C, respectively. Under drought-stressed conditions, the lowest CTDA (2.2˚C) was recorded by genotype ETBW8984, whereas the highest was showed by genotype ETBW8491. The highest CTDA under well-watered condition was recorded by genotype ETBW9088 (5.06˚C), followed by ETBW8491 (4.97˚C) and Tsehay (4.90˚C), while genotype ETBW8984 had the least (2.83˚C). The CTD at milking (CTDM), ranged from (1.95˚C) for genotype ETBW9449 to 3.68˚C for genotype Alidoro under drought-stressed condition, whereas it ranged from 2.13˚C for ETBW9441 to 4.12˚C for ETBW8491 under well-watered conditions. The highest CTD at dough (CTDD) under drought-stressed condition (2.92˚C) was recorded by genotype ETBW9088, while genotype ETBW9438 recorded the lowest (1.56˚C). The CTDD was highest for genotype ETBW8725 (3.11˚C) and lowest for genotype ETBW8944 (1.81˚C) under well-watered condition. The range in CTD at ripening (CTDR) under well-watered and drought-stressed conditions were 1.41˚C (ETBW8981) to 2.42 ETBW8725 and 1.15˚C (Menze) to 2.35˚C (ETBW9088), respectively.

Under well-watered condition, the highest SPAD readings at heading (SPADH), anthesis (SPADA), milking (SPADM), dough (SPADD) and ripening (SPADR) were recorded 58.5%, 60.0%, 55.0%, 51.0% and 44.9%, respectively for genotype Alidoro (Tables 3 and 4 and S5 Table). Correspondingly, the lowest SPADH (42.6%), SPADA (43.9%), SPADM (39.9%), SPADD (36.3%) and SPADR (31.4) values were observed for the genotype ETBW9441 under well-watered condition. The highest SPADH (52.4), SPADA (53.6) and SPADD (47.9) were recorded for the genotype Alidoro, and SPADR (41.90) for genotypes Alidoro and Tsehay under drought-stressed condition (Tables 3 and 4 and S4 Table). However, genotype Tsehay had the highest SPADM value of 50.4. The lowest SPADH (38.0), SPADA (39.1) and SPADM (36.8) values were recorded for genotype ETBW9441, and SPADD (33.9) and SPADR (29.2) were exhibited for genotypes ETBW9449 and ETBW8881 (Tables 6 and 7 and Appendix IV). The SPAD readings of the flag leaf of the wheat genotypes were in general lower under drought-stressed condition compared with well-watered condition across the phenological stages considered in this study. The genotypes Alidoro and Tsehay showed the highest SPAD values at all phenological stages under drought-stressed conditions.

The SPAD reading significantly reduced due to drought stress at heading, anthesis, milking, dough and ripening stages. The SPADH of 96 (49.0%) of genotypes were recorded above the mean values of standard checks, and 102 (52.0%) of genotypes were showed above the mean value across the genotypes under well-watered conditions. Under drought-stressed conditions, the SPADH reading of 90 (45.9%) of genotypes were recorded above the mean across the genotypes, whereas 91 (46.4%) of genotypes were showed above the mean value of standard checks. The mean percent reduction in SPAD reading at heading was 21.4% across genotypes showed significant declining trend due to drought stress (Tables 3 and 4 and S4 and S5 Tables. The 55 (28.1%) genotypes for SPDM, 61 (31.1%) for SPADD and 30 (15.3%) of genotypes for SPADR were recorded above the mean values of standard checks, and 102 (52.0%) of genotypes for SPADM, 103 (53.6%) for SPADD, and 60 (30.6%) for SPADR, 90 (45.9%) for SPDM, 51 (2%) for SPADD and (15.3%) SPADR were showed above the mean across the genotypes under well-watered condition. Under drought-stressed condition, 90 (45.9%) of genotypes for SPADA, 146 (74.4%) of genotypes for SPADM, 90 (45.9%) for SPADD, and 90 (45.9%) of genotypes for SPADR were recorded above the mean across the genotypes; whereas 71 (36.2%) of genotypes for SPADA, 93 (47.4%) for SPADM, 67 (34.2%) for SPADD, and 51 (26.0%) of genotypes for SPADR were showed above the mean of standard checks under drought-stressed condition. The mean percent reduction in SPAD at anthesis 16.5%, at milking 18.1%, at dough

**Table 6. Pearson's correlation coefficients of the association among fourteen physiological characters of 196 bread wheat genotypesevaluated under drought-stressed (below the diagonal) and well-watered (above the diagonal) conditions.**

|  | | Well-watered conditions | | | | | | | | | | | | | |
|---|---|---|---|---|---|---|---|---|---|---|---|---|---|---|---|
|  | Traits | CTDH | CTDA | CTDM | CTDD | CTDR | SPADH | SPADA | SPADM | SPADD | SPADR | RWC | ELWR | RWL | LMSI |
| Drought-stressed conditions | CTDH | **1.00** | 0.71** | 0.69** | 0.62** | 0.62** | 0.43** | 0.42** | 0.38** | 0.34** | 0.37** | 0.39** | 0.33** | -0.23* | 0.45** |
|  | CTDA | 0.83** | **1.00** | 0.64** | 0.63** | 0.55** | 0.48** | 0.45** | 0.44** | 0.40** | 0.45** | 0.49** | 0.40** | -0.28* | 0.49** |
|  | CTDM | 0.83** | 0.84** | **1.00** | 0.78** | 0.70** | 0.30** | 0.33** | 0.28** | 0.23* | 0.30** | 0.22* | 0.26* | -0.30** | 0.23* |
|  | CTDD | 0.79** | 0.81** | 0.91** | **1.00** | 0.78** | 0.34** | 0.34** | 0.33** | 0.29* | 0.34** | 0.25* | 0.28* | -0.35** | 0.28* |
|  | CTDR | 0.74** | 0.72** | 0.80** | 0.84** | **1.00** | 0.41** | 0.44** | 0.39** | 0.34** | 0.40** | 0.25* | 0.32** | -0.42** | 0.23* |
|  | SPADH | 0.32** | 0.39** | 0.34** | 0.35** | 0.49** | **1.00** | 0.93** | 0.95** | 0.91** | 0.88** | 0.56** | 0.38** | -0.38** | 0.69** |
|  | SPADA | 0.35** | 0.40** | 0.37** | 0.39** | 0.54** | 0.94** | **1.00** | 0.90** | 0.84** | 0.86** | 0.54** | 0.44** | -0.40** | 0.54** |
|  | SPADM | 0.30** | 0.37** | 0.34** | 0.35** | 0.50** | 0.95** | 0.97** | **1.00** | 0.97** | 0.89** | 0.46** | 0.30* | -0.39** | 0.67** |
|  | SPADD | 0.27* | 0.35** | 0.31** | 0.32** | 0.46** | 0.95** | 0.93** | 0.97** | **1.00** | 0.88** | 0.41** | 0.22* | -0.35** | 0.71** |
|  | SPADR | 0.32** | 0.37** | 0.37** | 0.39** | 0.56** | 0.87** | 0.88** | 0.89** | 0.89** | **1.00** | 0.51** | 0.37** | -0.41** | 0.59** |
|  | RWC | 0.60** | 0.53** | 0.53** | 0.51** | 0.52** | 0.39** | 0.40** | 0.38** | 0.36** | 0.34** | **1.00** | 0.83** | -0.24* | 0.55** |
|  | ELWR | 0.53** | 0.48** | 0.48** | 0.45** | 0.44** | 0.28* | 0.31** | 0.28* | 0.25* | 0.25* | 0.83** | **1.00** | -0.44** | 0.25* |
|  | RWL | -0.09ns | -0.17ns | -0.06ns | -0.08ns | -0.16ns | -0.45** | -0.39** | -0.43** | -0.45** | -0.36** | -0.20ns | -0.07ns | **1.00** | -0.27* |
|  | LMSI | 0.75** | 0.69** | 0.68** | 0.66** | 0.65** | 0.42** | 0.42** | 0.41** | 0.39** | 0.41** | 0.73** | 0.63** | -0.26* | **1.00** |

** = Significant at 1% level,

* = Significant at 5% level,

ns = Not significant, CTDH = canopy temperature depression at heading, CTDA = canopy temperature depression at anthesis, CTDM = canopy temperature depression at milking, CTDD = canopy temperature depression at dough, CTDR = canopy temperature depression at ripening, SPADH = SPAD heading, SPADA = SPAD anthesis, SPADM = SPAD milking, SPADD = SPAD dough, SPADR = SPAD ripening, RWC = Relative water content, ELWR = excised leaf water retention, RWL = Relative water loss, LMSI = leaf membrane stability index.

18.7% and at ripening 15.9% across genotypes showed significant declining trend due to drought stress (Tables 4 and 5 and S3 and S4 Tables).

The drought stress significantly resulted in declined mean values for the traits relative water content (RWC), excised leaf water retention (ELWR) and leaf membrane stability index (LMSI), however increased relative water loss (RWL) in almost all genotypes. The maximum RWC (76.8%) was maintained by the genotype by Alidoro, followed by ET13A2 (74.6%) and ETBW8491 (73.8) under well-watered conditions. Likewise, the highest RWC (67.6%) was kept by the genotype Alidoro, followed by ETBW172938 (65.25%) and Dinknesh (65.17%) under drought-stressed condition as indicated in Tables 4 and 5. The average relative water content (RWC) across the genotypes was 65.5% under well-watered condition and 63.12% under drought-stressed condition (Table 3). Averaged across genotypes, drought stress reduced RWC by 15.3%. The mean lowest RWC (56.78%) was recorded for genotype ETBW8945 under well-watered condition and RWC of 51.57% was for genotype Katar under drought-stressed condition (Tables 3 and 4). The mean percent reduction in RWC due to drought-stressed condition compared to well-watered condition across genotypes showed 15.8%.

Excised leaf water retention (ELWR) varied significantly from 49.5% recorded for genotype ETBW8945 to 65.4% recorded for genotype ETBW9088 under well-watered condition. Under drought-stressed condition, ELWR ranged from 43.6% recorded for genotype Digelu to 62.5% recorded for genotype ETBW172872. Under well-watered conditions, the highest excised leaf water retention (ELWR) (63.4%) was recorded in genotype ETBW9088, followed by ETBW172872 (63.3%) and ETBW8491 (62.5%), while under drought-stressed conditions, the

**Table 7. Eigen vectors of the first three principal components and the proportions and cumulative contributions of fourteen physiological traits of 196 bread wheat genotypes evaluated in four test environments under well-watered and drought-stressed conditions.**

| Traits | Drought-stressed condition | | | Well-watered condition | | | |
|---|---|---|---|---|---|---|---|
| | PC1 | PC2 | PC3 | Traits | PC1 | PC2 | PC3 |
| CTDH | 0.251 | -0.340 | 0.231 | CTDH | 0.248 | 0.251 | 0.377 |
| CTDA | 0.262 | -0.258 | 0.271 | CTDA | 0.235 | 0.230 | 0.452 |
| CTDM | 0.279 | 0.302 | -0.039 | CTDM | 0.299 | 0.084 | -0.246 |
| CTDD | 0.268 | -0.191 | 0.206 | CTDD | 0.241 | 0.207 | 0.410 |
| CTDR | 0.284 | -0.082 | 0.152 | CTDR | 0.283 | 0.138 | 0.245 |
| SPADH | 0.281 | 0.278 | -0.036 | SPADH | 0.302 | 0.084 | -0.242 |
| SPADA | 0.279 | 0.292 | -0.060 | SPADA | 0.302 | 0.057 | -0.252 |
| SPADM | 0.281 | 0.293 | -0.052 | SPADM | 0.299 | 0.072 | -0.267 |
| SPADD | 0.281 | 0.297 | -0.075 | SPADD | 0.300 | 0.066 | -0.250 |
| SPADR | 0.280 | 0.272 | -0.035 | SPADR | 0.300 | 0.045 | -0.214 |
| RWC | 0.254 | -0.288 | -0.467 | RWC | 0.264 | -0.16 | -0.014 |
| ELWR | 0.236 | -0.336 | -0.611 | ELWR | 0.248 | -0.23 | 0.042 |
| RWL | -0.228 | 0.082 | -0.443 | RWL | -0.181 | 0.629 | -0.172 |
| LMSI | 0.268 | -0.294 | 0.064 | LMSI | 0.202 | -0.56 | 0.179 |
| Eigen value | 10.66 | 1.563 | 1.002 | | 9.90 | 1.28 | 1.194 |
| Proportion of total variance (%) | 79.8 | 8.3 | 3.9 | | 70.7 | 9.1 | 8.5 |
| Cumulative variance (%) | 79.8 | 88.1 | 92.0 | | 70.7 | 79.8 | 88.3 |

CTDH = canopy temperature depression at heading, CTDA = canopy temperature de-pression at anthesis, CTDM = canopy temperature depression at milking, CTDD = canopy temperature depression at dough, CTDR = canopy temperature depression at ripening, SPADH = SPAD heading, SPADA = SPAD anthesis, SPADM = SPAD milking, SPADD = SPAD dough, SPADR = SPAD ripening, RWC = Relative water content, ELWR = excised leaf water retention, RWL = Relative water loss, and LMSI = leaf membrane stability index.

maximum ELWR (55.7%) was recorded in genotype ETBW172872, followed by ETBW8870 (55.6%) and Alidoro (55.1%). Under well-watered conditions, 28 (14.3%) of genotypes were above the mean values of standard checks, and 54 (27.6%) of genotypes were above the mean across the genotypes. The significant decline in ELWR (23.8%) was observed due to drought stress.

The highest relative water loss (RWL) (44.8%) was recorded in genotype ETBW8983, followed by ETBW9384 (44.7%) and ETBW9411 (43.8%), whereas the lowest percentage of RWL was observed in genotype ETBW172872 (31.7%), followed by genotypes ETBW9088 (32.5%) and ETBW172936 (32.7%) under well-watered condition (Tables 3 and 4 and S5 Table). In this study, the maximum RWL (40.6%) was recorded in genotype ETBW8261, followed by ETBW8983 (40.5%) and ETBW9441 (40.5%), while the minimum RWL (30.1%) in genotype Alidoro, followed by Abola (31.9%) and ET13A2 (32.4%) under drought-stressed conditions (Tables 3 and 4). Under well-watered conditions, RWL of 28 (14.3%) of genotypes were recorded above the mean values of standard checks, and 54 (27.6%) of genotypes were showed above the mean across the genotypes (S4 Table). Under drought-stressed conditions, RWL of 106 (54.1%) of genotypes were recorded above the mean across the genotype; whereas 135 (68.9%) of genotypes were showed the above the mean values of standard checks (S4 Table). The mean percent reduction in RWL (23.8%) across genotypes showed significant declining trend due to drought stress as compared to well-watered condition (Tables 3 and 4).

Leaf membrane stability index (LMSI) was reduced under drought-stressed condition relative to well-watered condition. Under drought-stressed condition, LMSI ranged from 37.8% to

69.1%, with a mean of 51.9%, while it ranged from 55.0% to 77.2%, with a mean of 66.0% under well-watered condition. Genotype Alidoro had the highest LMSI (69.1%), followed by genotypes ETBW8491 (66.4%) and ETBW8870 (65.3%), while genotype ETBW9441 (37.8%) had the lowest LMSI followed by genotypes Honqolo (37.8%) and Galema (37.9%) under drought-stressed conditions (Table 4 and S4 Table). Under well-watered condition, genotype ETBW9088 had the highest LMSI (77.2%) followed by genotypes ETBW8491 (76.9%) and ETBW8870 (76.0%), while genotypes ETBW8827 (55.00%), ETBW8984 (55.6%) and ETBW9402 (55.6%) had the lowest LMSI (Tables 3 and 4 and S4 Table). The LMSI of 46 (23.5%) of genotypes were recorded above the mean values of standard checks, and 105 (53.6%) of genotypes were showed above the mean across the genotypes under well-watered conditions. Under drought-stressed conditions, LMSI of 112 (57.1%) of genotypes were recorded above the mean across the genotype; whereas 51 (26.0%) of genotypes were showed above the mean values of standard checks. The mean percent reduction in LMSI (20.2%) across genotypes showed significant declining trend due to drought stress (Tables 3 and 4 and S4 and S5 Tables). Genotypes Alidoro, ETBW8491 and ETBW8870 showed low percentage decline in LMSI in the leaves.

**3.3.1 Relationships among physiological traits.** The estimates of correlation coefficients indicating the degree of associations among measured physiological characters under drought-stressed and well-watered conditions are presented in Table 6. In the present study, all canopy temperature depression traits (CTDH, CTDA, CTDM, CTDD and CTDR) were positively and significantly correlated with one another, and with other traits under both water regimes, but inverse relationship with RWL. However, the relationships between these traits and RWL were negative, and significant only under well-watered condition. Similarly, flag leaf chlorophyll content (SPAD-502 value) measured at the different stages (SPADH, SPADA, SPADM, SPADD and SPADR) had significant and positive associations with one another and with other characters under drought-stressed and well-watered conditions, except RWL. On the other hand, the relationships of the flag leaf chlorophyll content traits with RWL under both water regimes were negative and highly significant. The SPAD values were more strongly affected by location and drought-stress.

Relative water content (RWC) was significantly and positively associated with ELWR and LMSI under both well-watered and drought-stressed conditions. The rela-tionships between RWC and RWL under both water regimes were negative, and significant only under well-watered condition. Excised leaf water retention (ELWR) had negative and significant correlation with relative water loss (RWL) under well-watered condition, but the correlation was not significant under drought stress, indicating that drought tolerant genotypes had the lower RWL and higher ELWR. However, the associations of ELWR with LMSI were positive and significant under both drought-stressed and well-watered conditions. Relative water loss (RWL) was negatively and significantly associated with LMSI under drought-stressed and well-watered conditions.

**3.3.2. Principal component analysis.** Principal component analysis (PCA) revealed that under drought-stressed condition the first three principal components (PC1, PC2 and PC3) cumulatively accounted for about (92.0%) of the total variation among the genotypes, with PC1 contributing 79.8% (Table 7). The contributions of the second and third principal components (PC2 and PC3) to the total variation among the genotypes were relatively very small (8.3% and 3.9%, respectively). The individual effects of most traits to the variation in PC1 were small ranging from -0.228 for RWL to 0.284 for CTDR, with all the traits, except RWL making positive contributions. The major positive contributors to PC2 were CTDM, SPADH, SPADA, SPADM, SPADD and SPADR, while PC3 had major positive loadings from CTDA, CTDH, CTDD and CTDR.

Under well-watered condition, the first three principal components cumulatively accounted for 88.3% of the total variation among the genotypes, out of which PC1 contributed 70.7% (Table 7). The contributions of the second and third principal components (PC2 and PC3) to the total variation among the genotypes were 9.1% and 8.5%, respectively. The individual effects of most traits to the variation in PC1 were small ranging from -0.181 for RWL to 0.302 for SPADH and SPADA. All the traits, except RWL made positive contributions to PC1. Similarly, all the traits, except RWC, ELWR and LMSI contributed positively to PC2. The third principal component (PC3) had positive loadings from CTDA, CTDH and CTDD, CTDR, ELWR and LMSI.

The genotype by trait biplot of the first and second principal components (PC1 and PC2) for the well-watered condition is presented in Fig 3. Under well-watered conditions, most of

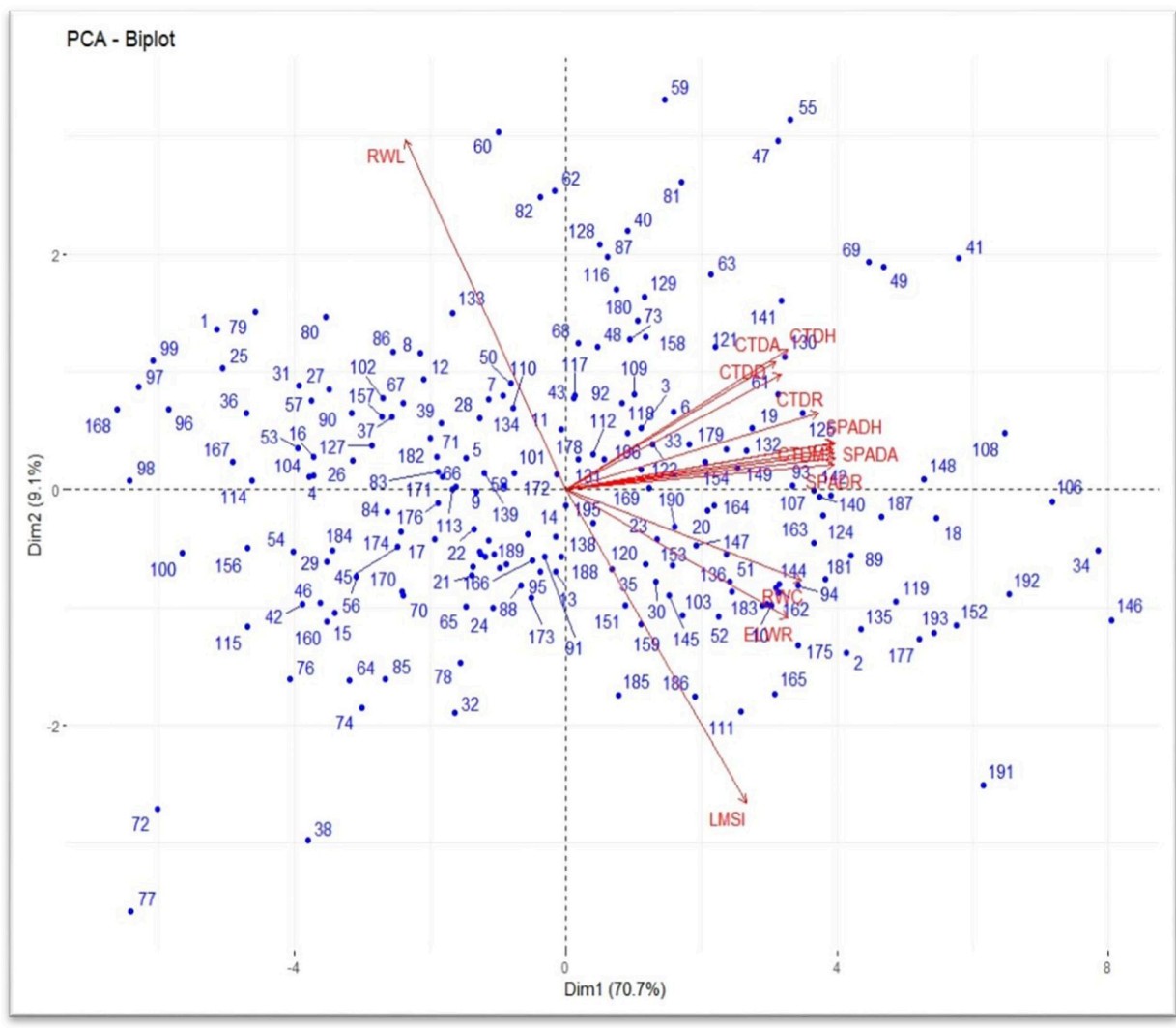

**Fig 3. PCA biplot of 14 physiological traits of 196 genotypes of bread wheat under well-watered condition.** CTDH = canopy temperature depression at heading, CTDA = canopy temperature depression at anthesis, CTDM = canopy temperature depression at milking, CTDD = canopy temperature depression at dough, CTDR = canopy temperature depression at ripening, SPADH = SPAD heading, SPADA = SPAD anthesis, SPADM = SPAD milking, SPADD = SPAD dough, SPADR = SPAD ripening, RWC = Relative water content, ELWR = excised leaf water retention, RWL = Relative water loss and LMSI = leaf membrane stability index.

the genotypes and traits were loaded on the positive axis side of PC1 (Fig 3). Among the traits, CTDH, CTDA, CTDM, CTDD, CTDR, SPADH, SPADA, SPADM, SPADD and SPADR were associated positively with each other and most contributing traits on the first principal component. Although the magnitude was relatively low, ELWR and RWC were positively correlated with each other on PC1, but negatively related with RWL on PC2. Furthermore, RWL is highly negatively related with LMSI on extremes of inverse vector directions and essentially contributed to overall diversity (Fig 3). Genotypes Abola, Danda'a, Alidoro, ET-13A2, Tsehay, ETBW8817, ETBW8816, ETBW8831, ETBW9402, ETBW8394, ETBW9422, ETBW8489, ETBW8725, ETBW8996, ETBW9087, ETBW9139, ETBW9470, ETBW9169, ETBW8583 and ETBW172996 were distributed on the positive side of the PC1 along with traits CTDH, CTDA, CTDM, CTDD, CTDR, SPADH, SPADA, SPADM, SPADD and SPADR, therefore considered as superior genotypes in one or more of the traits. The bread wheat genotypes Doddota, Galema, ETBW9449, ETBW9435, ETBW9440, ETBW9441, ETBW9414, ETBW8981, ETBW8797 and ETBW9176 were scattered on the negative side of the PC2 along with traits RWC, ELWR, RWL and LMSI, hence they can be considered as the drought susceptible genotypes. Other remaining genotypes are considered as neither tolerant nor susceptible in terms of the performance under well-watered condition.

On the biplot of PC1 and PC2 under drought-stressed condition, most of the genotypes and traits were observed to be concentrated on the positive side of PC1 (Fig 4). Traits CTDM, SPADH, SPADA, SPADM, SPADD and SPADR were positively associated with each other and were the most contributing traits to PC1. This indicated that the traits were major contributors to the total variation under conditions of drought stress. Traits CTDR, CTDD, CTDA, ELWR, CTDH and RWC were positively associated with each other on PC1 with relatively low magnitude. Genotypes such as Alidoro, ET-13A2, Kingbird, Tsehay, ETBW8816, ETBW9027, ETBW9402, ETBW8394 and ETBW8725 along with traits CTDM, SPADH, SPADA, SPADM, SPADD and SPADR were distributed to the positive side of PC1, therefore the genotypes were considered as the drought tolerant. On the other side, genotypes like Ogolcho, Galil, Doddota, Galema, ETBW9449, ETBW9435, ETBW9441, ETBW8983, ETBW8944, ETBW8984, ETBW8945, ETBW8981, ETBW8797, ETBW9176 and ETBW8585 were scattered to the negative side of the axis, therefore they can be considered as the drought susceptible genotypes.

## 4. Discussion

### 4.1 Morphological traits

**4.1.1 Flag leaf size.** The photosynthetic performance of a plant during drought stress has been reported to depend on the morphological properties of flag leaf [50]). Findings from the study reported by [51] indicated that drought tolerant wheat genotypes had smaller, narrower, more erect, but thicker waxy leaves with higher photosynthetic activity, increased relative water content and low water loss under drought-stressed condition. Also, Kumar *et al.* [52] reported that genotypes with narrow sized leaves compared to large sized leaves had better drought tolerance than other genotypes under drought-stressed conditions by reducing the leaf water loss. In this study, bread wheat genotypes with large flag leaves had droopy and weakly-rolled to no-rolled leaves than narrow-leaved genotypes; however, the genotypes with small flag leaves had erect flag leaves under both well-watered and drought-stressed conditions. This finding is consistent with the report of [53–55], who reported that relatively smaller flag leaf size and erect flag leaf angle genotypes reduce water loss and can improve light absorption, which improves photosynthesis.

**4.1.2 Flag leaf angle.** Genotypes with erect leaf angle and narrow leaf size have been associated with higher photosynthesis in cereal crops, such as barley, wheat, sorghum, and oats

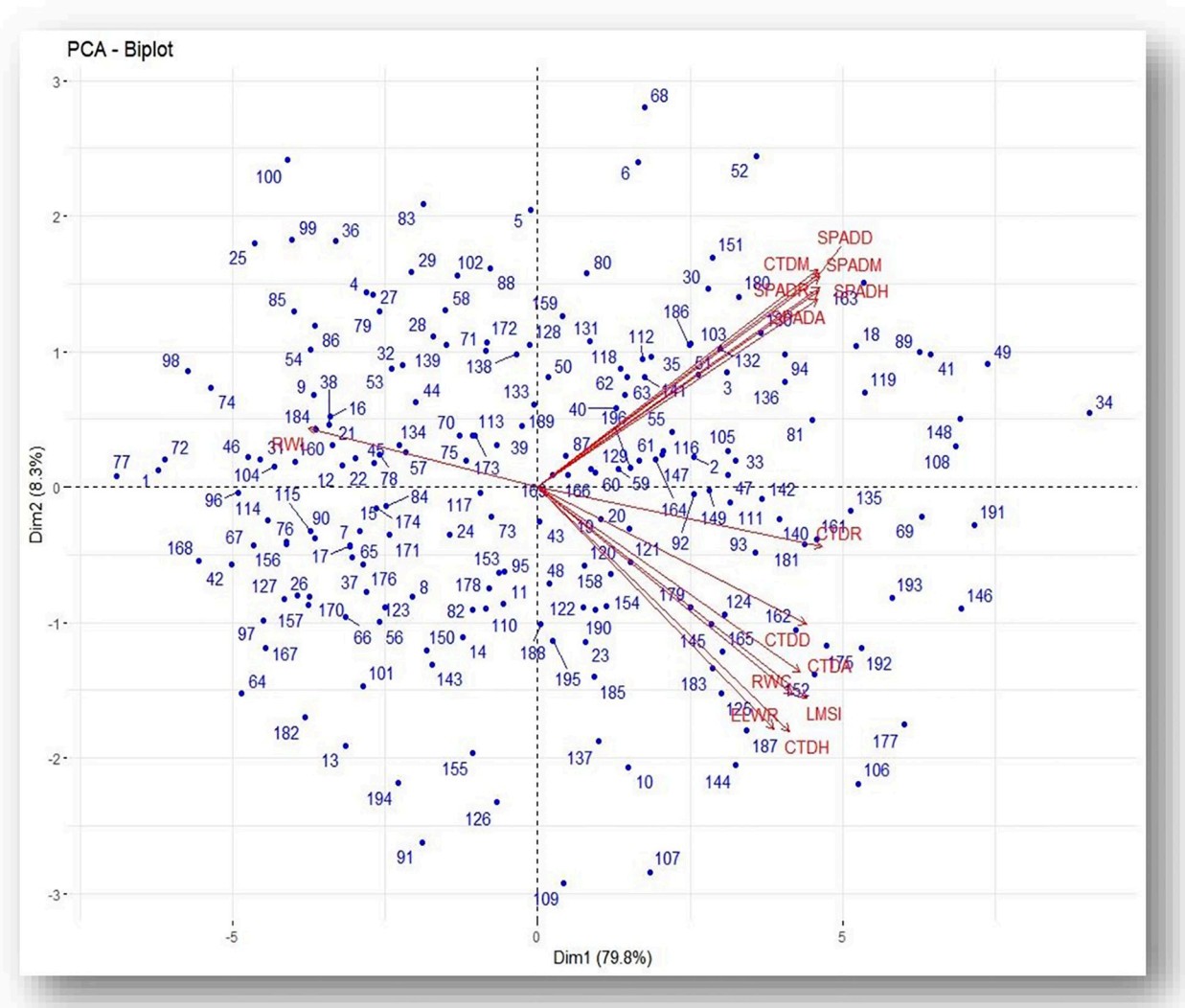

**Fig 4. PCA biplot of 14physiological traits of 196 genotypes of bread wheat under drought-stressed condition.** CTDH = canopy temperature depression at heading, CTDA = canopy temperature depression at anthesis, CTDM = canopy temperature depression at milking, CTDD = canopy temperature depression at dough, CTDR = canopy temperature depression at ripening, SPADH = SPAD heading, SPADA = SPAD anthesis, SPADM = SPAD milking, SPADD = SPAD dough, SPADR = SPAD ripening, RWC = Relative water content, ELWR = excised leaf water retention, RWL = Relative water loss and LMSI = leaf membrane stability index.

under drought-stressed conditions [56]. The advantages of erect leaves in increasing the growth rate [57] and improving radiation use efficiency in high radiation environments under drought-stressed conditions had been demonstrated in wheat [58]. Additionally, narrow and upright leaves had greater leaf area exposed to sunlight, and increase photosynthetic rate in wheat genotypes [27]. Breeding goals for wheat improvement should therefore include the reduction of leaf angle to increase plant density, improve light interception efficiency, and increase chlorophyll concentration [56].

**4.1.3 Flag leaf rolling.** Leaf rolling has the potential to improve the efficiency of breeding for drought tolerance in wheat [59]. Since leaf rolling is associated with water loss in cultivated wheat, it is a potential proxy trait for screening wheat genotypes up to a certain level of water

loss from the leaves [38]. Rebetzke *et al.* [60] had earlier reported genotypic differences for flag leaf rolling in wheat. But studies related to the genotypic difference of leaf rolling are rare in wheat [61]. This finding is consistent with what was reported by [62] that wheat genotypes which exhibited strong or full leaf rolling had more water use efficiency than genotypes that exhibited no leaf rolling under drought-stressed and non-stressed conditions. Ali *et al.* [63] had reported that wheat leaf rolling and its association with leaf surface wetness helps the plant in water retention. In wheat, leaf rolling facilitates efficient photosynthetic activities and enhances accumulation of dry matter [64, 65], lowers leaf surface temperature and reduces water loss by slowing down transpiration [66].

**4.1.4 Waxiness of leaf.** In wheat breeding, waxiness is used to select drought-tolerant genotypes. The content of wax in the leaf is a shoot morphological trait known to influence water loss from the cuticular surface and is one of the features used to select for drought tolerance [67, 68]. The study of [69] reported that wheat varieties with high waxiness had reduced water loss from plant surfaces. Similarly [70], reported that waxy cuticle would be advantageous in plant adaptation to drought as it reduces plant water loss. The improved drought tolerance of the genotypes with high waxiness results from increased reflection of solar radiation and reduced cuticular permeability to water loss and consequently maintaining higher leaf water potential [71]. Under moisture-stressed conditions, leaf waxiness or glaucousness has been shown to be associated with cooler canopies [72].

**4.1.5 Disease severity score.** The bread wheat genotypes tolerant to drought stress showed low disease severity. Low SPAD with high AUDPC values for the genotypes suggests that the chlorophyll content of the genotypes decreases as the disease severity increases [73]. High disease severity cause rapid necrosis of the flag leaf, thereby reducing chlorophyll content. Resistant to disease severity genotypes having higher SPAD value were showed tolerant to drought stress, while susceptible genotypes with lower SPAD value were revealed susceptible to drought stress. Similar results were obtained by [74].

## 4.2 Physiological traits

**4.2.1 Analysis of variance under drought-stressed and well-watered conditions.** For all of the physiological traits studied, the analysis of variance (Tables 1 and 2) revealed substantial variations across the bread wheat genotypes, indicating the presence of significant genetic diversity that may be used to increase drought tolerance. Presence of genetic variability among the test genotypes for traits related to stress tolerance is of paramount importance for successful breeding aimed to develop cultivars adapted to a range of stress environments [75–77]. In the current study, there was a ranking in the physiological traits of the test genotypes under drought-stressed and well-watered conditions, suggesting differential genetic responses to changes in water availability to identify superior genotypes and genotypes adapted to specific or broad adaptation. Heritability in broad sense ($H^2BS$) values ranged from 43.28% for RWL to 85.75% for the trait SPADA under well-watered conditions while a range of 43.02% for trait RWL to 87.29% for the trait LMSI was observed under drought-stressed conditions. Across the water regimes, the values of heritability in broad sense ranged between 43.02% and 87.29%. CTDA, SPADA and SPADM had high heritability estimates (above 79%) under both drought-stressed and well-watered conditions. RWL had the lowest broad sense heritability (<45%) under both drought-stressed and well-watered conditions.

The results of this study in Tables 1 and 2 showed the high value of broad sense heritability 87.29% for the trait LMSI indicated that LMSI can be utilized as a selection criterion for breeding cultivars under drought stress conditions. Variation in heritability of the traits would increase their importance in the improvement of drought-tolerance [78]. High heritability

estimates along with high genetic advance, signifying that these traits were under the influence of additive genes, which favours their improvement via direct selection, even under drought-stressed conditions [79–81].

**4.2.2 Physiological response of the genotypes.** Production of bread wheat is primarily limited by drought stress, and changing climatic conditions are making the situation even the worst. In the current study, the physiological response of the genotypes to drought stress tolerance in 196 bread wheat genotypes were evaluated on the basis of physiological traits. Drought stress causes significant decreases in canopy temperature depression (CTD), leaf chlorophyll content, relative water content (RWC), excised leaf water retention (ELWR), and leaf membrane stability index (LMSI) in all the studied bread wheat genotypes; but increases relative water loss (RWL). Canopy temperature depression (CTD), defined as deviation of plant canopies temperature from the ambient temperature, has been recognized as the key trait for comparison of genotypes response to low water use [51]. Selecting genotypes for high CTD under both drought-stressed and well-watered conditions allow genotypes to maintain ranks for high CTD since the same genotypes will be expected to perform well in either situation. Therefore, canopy temperature depression used as selection criterion for breeders when selecting for drought-tolerant genotypes. The comparatively lower CTD detected in drought-stressed crop plants indicates the potential for absorbing higher soil moisture which assists the plants to maintain optimal water status [82].

Based on SPAD measurement values, selection of genotypes should be done with great care for development of drought tolerant genotypes. Mean SPAD measurement values at booting, heading, anthesis and physiological maturity have been used successfully by many researchers in the screening and selection of drought-tolerant wheat cultivars [83]. The wheat varieties that are susceptible to drought stress showed significant decline in SPAD values, while tolerant varieties revealed higher SPAD readings than susceptible ones [84]. In our research, the reduction in SPAD reading values varies from 15.9% for SPADD to 18.7% for SPADM were observed across the genotypes. Similarly, a 13 to 15% reduction in chlorophyll content was observed in wheat varieties due to limited water supply [85]. Drought stress prevents different physiological processes in plants, which ultimately leads to a reduction in the growth and yield traits of plants. Extensive drought stress inhibits photosynthesis due to a change in chlorophyll content. Pour-aboughadareh *et al.* [86] reported that there was a decrease in leaf chlorophyll content due to drought stress in durum wheat. Similarly, the present study showed a reduction in the chlorophyll content of all the studied bread wheat genotypes under drought stress compared to well-watered conditions. Wheat varieties exhibited differences in SPAD-chlorophyll content and higher SPAD-chlorophyll value was observed under well-watered conditions while it was significantly reduced under drought conditions [87].

The leaf relative water content (RWC) indicates the ability of plants to keep their water status adequate to sustain water stress. The findings of our research show that the RWC dropped due to drought-stressed conditions in all the studied bread wheat genotypes at different levels. Moreover, it has been shown that the RWC is negatively correlated with the relative water loss, and the larger the water loss from leaves revealed by reduced the RWC [88]. Differences in RWC among genotypes may be due to increased root depth and diversity in genetic potential for water uptake from the rhizosphere [89]. Our research findings were in accordance with the results of [90] who also reported comparatively most genotypes showed higher RWC under well-watered condition than drought-stressed condition. The existence of genotypic variations among bread wheat genotypes in RWC have been reported in response to drought stress (Belay et al., 2021). The decrease in RWC under drought-stressed conditions might be due to decreased crop vigour [19]. The findings of our works were in agreement with the

results obtained by [31], who observed that bread wheat genotypes treated to conditions of drought stress have greatly reduced relative water content. Almeselmani *et al.* and Kocheva *et al.* [83, 91] reported that drought stress sufficiently decreases the relative water content, osmotic potential and nutrient uptake of plants, which finally results in a low leaf turgor pressure and reduced metabolic activities. The genotypes that maintained more water content in their leaves and low water loss from leaves under well-watered and drought-stressed conditions can tolerate drought stress. In this research, the genotypes Alidoro, Abola and ET13A2 showed the lowest RWL under drought-stressed condition compared to well-watered condition, which indicated that they are better drought-tolerant than other genotypes. Khan *et al.* [90] reported that bread wheat genotypes which showed lower RWL are drought-tolerant, which confirm the findings of this study.

Genotypes showed high ELWR under both drought-stressed and well-watered conditions tend to have higher potential to preserve water balance in leaves, and used to select drought-tolerant genotypes. Previously, researchers have been used the high mean values of ELWR in selection and screening of drought-tolerant bread wheat cultivars for utilisation in breeding programmes [54, 92]. Under drought-stressed conditions, leaf membrane stability is considered as an important factor to tolerate drought stress during selection of cultivars [93]. The drought-tolerant wheat genotypes possess stable leaf membrane and can maintain its integrity [19]. The previous findings reported by [94, 95] about some wheat varieties showed a lower percentage reduction in LMSI and better water retention in the leaves under drought-stressed conditions. In our results, the leaf membrane stability index was highly affected under drought-stressed conditions, leading to relative water loss. Our findings are supported by the previous observation of [96, 97] who have reported the correlation of electrolyte leakage with drought stress tolerance. Genotypes that showed high LMSI, RWC and ELWR, CTD, chlorophyll content and lower RWL contributed to screening drought-tolerant genotypes of bread wheat for drought-prone areas of Ethiopia. Consequently, the identified genotypes and traits could be used in future breeding programmes targeting for drought tolerance.

**4.2.3 Relationships among physiological traits.** The term "canopy temperature depression" refers to the difference between the air temperature and canopy temperature. Under drought-stressed conditions, an increase in water transpiration can result in a decrease in plant surface temperature, and vice versa [98]. The increase in respiration and decrease in transpiration brought on by stomatal closure during drought stress are likely the causes of the CTD decline [31]. Additionally, CTD has the advantage of being a non-destructive method for assessing changes in stomatal conductance under drought-stressed conditions, and it has been recognized as a crucial indication of water status in crop plants [99]. In this research, the CTD at heading, anthesis, milking, dough and ripening showed a high significant and positive correlation with each other, SPAD reading values, RWC, ELWR and LMSI. CTD and other traits showed that CTD has been used for selection criteria in breeding programmes. The positive and significant association of CTD with SPAD reading chlorophyll content indicated that an increase in one of the traits would lead to a linear increase in the other traits [100]. The flag-leaf SPAD reading values at the heading, anthesis, milking, dough and ripening stages showed a high significant positive association with each other and with CTD under drought-stressed and well-watered conditions. Significant correlation between SPAD reading values chlorophyll and CTD showed that canopy temperature depression and chlorophyll content can play effective role in identifying drought tolerant genotypes [101]. Drought stress tolerant wheat varieties had earlier been shown by [102, 103] to have high chlorophyll contents than drought stress sensitive genotypes. The results of this study indicated that genotypes with the higher RWC, ELWR, and LMSI were

characterized by the lower RWL. The RWC showed high significant and positive correlation with CTD and SPAD reading values; however it showed significant negative association with RWL [90]. In this study, the significant positive correlation between canopy temperature depression and chlorophyll content in bread wheat genotypes under drought-stressed conditions suggests that a higher canopy depression value is dependent on higher levels of chlorophyll content (Table 7). The significant positive correlation values of canopy temperature depression with LMSI, RWC and ELWR under both drought-stressed and well-watered conditions, suggests that transpiration from the leaves causes increase in CTD which is dependent on healthy leaf traits, i.e. stability of leaf membrane, high relative water content and high excised leaf water retention which can only be achieved by the tolerant varieties under drought stress [19].

When the soil moisture declines, crops water uptake by the roots becomes more difficult, and consequently RWC decreases [104]. Kocheva and Kartseva [91] confirmed that the greater water loss linked to the greater leaf membrane injure and decline RWC. Geravandia *et al*. [35] showed that higher LMSI, RWC, ELWR, and lower RWL could be considered as reliable indicators in bread wheat genotypes screening for drought tolerance. Thus, wheat genotypes tolerant to drought maintained higher CTD, chlorophyll content, and RWC than drought sensitive genotypes.

## 5. Conclusion

This study was conducted at field and greenhouse environments in two water regimes in order to evaluate the consistency of morphological and physiological traits for drought-tolerant bread wheat genotypes screening purposes. From the results of this study, drought-tolerant genotypes had narrow flag leaved, erect leaf angle, fully rolled flag leaf, waxy leaved and resistant to highly resistant genotypes to disease under drought-stressed condition. The separate and combined analyses of the studied genotypes showed highly significant differences for the all physiological traits, suggesting the existence of wide variability among genotypes. Based on best performance in low RWL under drought-stressed condition, genotypes Alidoro, BW172938, BW8491, Bolo, BW172872, BW9088, BW8870, BW172936, ET13A2, Abola, BW8492, BW8303, BW8725, and BW9027 were considered as drought-tolerant genotypes. The findings of this study revealed that drought tolerant genotypes had the maximum CTDM, LMSI, RWL, SPADA, SPADH, SPADM and the lower RWL in comparison to drought sensitive genotypes. In the present study, all canopy temperature depression traits (CTDH, CTDA, CTDM, CTDD and CTDR) and flag leaf chlorophyll content (SPAD502 value) measured at the different stages (SPADH, SPADA, SPADM, SPADD and SPADR) were positively and highly significantly correlated with one another, and with other traits under both water regimes, except RWL. RWL showed that highly significant and negative association with SPADH, SPADA, SPADM, SPADD, SPADR and LMSI under both water regimes. The trait CTDH, CTDA, CTDM, CTDD, CTDR and RWC had not-significant and negative correlation with RWL under drought-stress condition. The PCA biplots results revealed that Alidoro, ET-13A2, Kingbird, Tsehay, ETBW8816, ETBW9027, ETBW9402, ETBW8394 and ETBW8725 along with traits CTDD, CTDM, CTDR, SPADA, SPADM, SPADD and SPADR were distributed in the positive side of first PC under both water regimes, therefore considered as the potential drought tolerant genotypes. Consequently, the identified genotypes and traits could be used in future breeding programmes targeting for drought tolerance. Finally, high LMSI, RWC and ELWR, the higher CTD, higher leaf chlorophyll content, and lower RWL contributed to screening drought-tolerant genotypes of bread wheat for drought-prone areas of Ethiopia.

## Supporting information

**S1 Table. Average monthly temperature, relative humidity and precipitation at the research site during the experiment.**
(DOCX)

**S2 Table. The genotype name, pedigree and origin details of the eight standard checks.**
(DOCX)

**S3 Table. List of genotypes used in the study.**
(DOCX)

**S4 Table. Mean values of the 14 physiological traits for 196 bread wheat genotypes under drought-stressed conditions.**
(DOCX)

**S5 Table. Mean values of the 14 physiological traits for 196 bread wheat genotypes under well-watered conditions.**
(DOCX)

**S6 Table. The tested genotypes flag leaf size was grouped based on its length and width category as described in the following table.**
(DOCX)

**S1 Fig. Box plots of physiological traits under WWC and DSC.**
(TIFF)

**S2 Fig. Flag leaf size.**
(TIFF)

**S3 Fig. Cluster Dendrogram under DSCs.**
(JPEG)

**S4 Fig. Cluster phylogenic under DSCs.**
(JPEG)

**S5 Fig. Correlation pairs under DSCs.**
(JPEG)

**S6 Fig. Correlation pie plot under DSCs.**
(JPEG)

**S7 Fig. Principal component analysis under DSCs.**
(JPEG)

**S8 Fig. Principal component analysis under WWCs.**
(JPEG)

**S9 Fig. Correlation coefficient analysis pie under WWCs.**
(JPEG)

**S10 Fig. Correlation pairs plot under WWCs.**
(JPEG)

**S11 Fig. Dendrogram of physiological traits under WWCs.**
(TIFF)

**S12 Fig. Phylogenic tree under WWCs.**
(JPEG)

**S1 File. Data of physiological traits under drought-stressed conditions.**
(XLSX)

**S2 File. Data of physiological traits under well-watered conditions.**
(XLSX)

**S3 File. Maximum, Minimum and mean values of physiological traits under WWCs and DSCs.**
(XLSX)

**S4 File. Morphological traits under drought-stressed conditions.**
(XLSX)

**S5 File. Morphological traits under WWCs.**
(XLSX)

## Acknowledgments

We would like to thank Zerihun Tadesse and Habte Zegeye from Kulumssa Agricultural Research Center, Ethiopia for providing the genetic materials used for the experiments. We would also thank Hailu Lire Wachemo from ILRI, Muluken Birarra from Addis Ababa Institute of Biotechnology and A. Raouf Sayadi from IITA for their kind assistance with statistical analyses.

## Author Contributions

**Conceptualization:** Birhanu Mecha Sewore.

**Data curation:** Birhanu Mecha Sewore.

**Formal analysis:** Birhanu Mecha Sewore.

**Funding acquisition:** Birhanu Mecha Sewore.

**Investigation:** Birhanu Mecha Sewore.

**Methodology:** Birhanu Mecha Sewore.

**Project administration:** Birhanu Mecha Sewore, Ayodeji Abe.

**Resources:** Birhanu Mecha Sewore, Mandefro Nigussie.

**Software:** Birhanu Mecha Sewore.

**Supervision:** Ayodeji Abe, Mandefro Nigussie.

**Validation:** Ayodeji Abe, Mandefro Nigussie.

**Visualization:** Ayodeji Abe, Mandefro Nigussie.

**Writing – original draft:** Birhanu Mecha Sewore.

**Writing – review & editing:** Birhanu Mecha Sewore, Ayodeji Abe, Mandefro Nigussie.

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
