## [Decision Letter · Decision Letter 0]

8 Dec 2022

PONE-D-22-30476Evaluation of Bread Wheat <triticum aestivum="" l.=""> Genotypes for Drought Tolerance using Morpho-physiological Traits under Drought-stressed and Well-watered Conditions</triticum>PLOS ONE

Dear Dr. Sewore,

Thank you for submitting your manuscript to PLOS ONE. After careful consideration, we feel that it has merit but does not fully meet PLOS ONE’s publication criteria as it currently stands. Therefore, we invite you to submit a revised version of the manuscript that addresses the points raised during the review process. Especially consider the suggestions made by reviewers regarding statistical analysis which will improve the quality of the data being presented. Please submit your revised manuscript by Jan 22 2023 11:59PM. If you will need more time than this to complete your revisions, please reply to this message or contact the journal office at plosone@plos.org. Please include the following items when submitting your revised manuscript:A rebuttal letter that responds to each point raised by the academic editor and reviewer(s). You should upload this letter as a separate file labeled 'Response to Reviewers'.A marked-up copy of your manuscript that highlights changes made to the original version. You should upload this as a separate file labeled 'Revised Manuscript with Track Changes'.An unmarked version of your revised paper without tracked changes. You should upload this as a separate file labeled 'Manuscript'.

We look forward to receiving your revised manuscript.

Kind regards,

Rattan Singh Yadav, PhD

Academic Editor

PLOS ONE

Journal Requirements:

"The funders had no role in the design of the study; in the collection, analyses, or interpretation of data; in the writing of the manuscript, or in the decision to publish the results."

"Author name: Birhanu Mecha

Grant number: no

African Union funded stipend, and field and lab research costs.

6. Please amend either the abstract on the online submission form (via Edit Submission) or the abstract in the manuscript so that they are identical.

**Additional Editor Comments:**

Your manuscript is interesting, however it needs revising. Please can you consider reviewers comments and resubmit for further considerations.

Reviewers' comments:

Reviewer's Responses to Questions

**Comments to the Author**

1. Is the manuscript technically sound, and do the data support the conclusions?

Reviewer #1: No

Reviewer #2: Partly

2. Has the statistical analysis been performed appropriately and rigorously? 

Reviewer #1: No

Reviewer #2: Yes

3. Have the authors made all data underlying the findings in their manuscript fully available?

Reviewer #1: Yes

Reviewer #2: No

4. Is the manuscript presented in an intelligible fashion and written in standard English?

Reviewer #1: Yes

Reviewer #2: No

5. Review Comments to the Author

Reviewer #1: The manuscript by Sewore et al. describes evaluation of 196 wheat genotypes under drought stress and optimum irrigation conditions in Ethiopia. The study is important and is of significant to Ethiopian breeders and to wheat scientists working on improving drought tolerance in wheat. However, there are many issues with the analysis and the presentation of results. Also, there are some fundamental errors, some of which can’t be improved just by revision.

1. Measurement of heritability’s of traits is missing.

2. Authors should add a figure of box plots of traits under optimum and drought stress conditions and use error bars.

3. Plant height and flowering time/days to heading are important traits to measure in drought stress trials. Is there any specific reason for not measuring these in the study?

4. BLUPs should have been calculated instead of just means in field experiments for all traits.

5. Authors should have created UPGMA trees to understand relationships among genotypes under both conditions separately and combined and should have discussed these relationships

6. If the objective is to characterize genotypes under drought and optimum irrigation conditions, why disease resistance scores were taken? What is the significance of adding disease resistance scores is not clear.

7. The Results and Discussion is poorly written and lacks synthesis

Reviewer #2: I have completed the review of this manuscript describing the evaluation of bread wheat genotypes under drought stress and well-watered conditions Significant revisions should be made to the submitted manuscript before it could be considered acceptable for publication. Please address the specific points noted below in a revised manuscript.

Line no 226: what range used for size grouping give in details? Because all should not have the same size

Line no 228: It seems that 22 genotypes with very large leaf size categorized into 56 genotypes under stress condition? give details of which genotypes leaf size reduced and level of change in leaf size in percentage. it will help to identify the better genotypes.

Line no 229: Author only gave graphical representation, give leaf pictures of also.

Line no 235-237: Part of flag leaf angle

Line no 241: Again, give leaf figure showing different leaf angles. What leaf to stem angle used for three different categories?

Line no 242-244: According to the results under drought condition number of genotypes for both droopy and erect were almost same (51 and 48) but under well-watered condition genotypes with droopy leaf angel were around 2.5 times more than number of genotypes with erect leaf angle.

Line no 244: As per study drooping occurs due to two main reasons: water stress and over watering

In this study drooping genotypes number should be increased under stress condition relative to well water conditions. results reveal that over watering was there.

Line no 246-248: Early leaf dropping is a good example of drought avoidance mechanism, why author not considered it for drought resistance scoring.

Line no 250-251: Yes, true in case of yield but if you are handling with drought stress

Line no 253-255: Again, under stress condition number of genotypes with leaf rolling is least, however, increased under well-watered condition. Authors give data in a random order. Always represent data in ascending or descending order. Write manuscript in a scientific way, under stress condition start results with data for no flag leaf rolling however, under well-watered condition started with flag leaf rolling.

PCA: Discuss grouping of genotypes on the basis of there pedigree. Genotypes within the group have same pedigree?

Fig 1 and 2; number in graphs overlapping, should use small font size.

Appendix C: give details of genotypes, their pedigree etc.

6. PLOS authors have the option to publish the peer review history of their article (what does this mean?). If published, this will include your full peer review and any attached files.

Reviewer #1: No

Reviewer #2: No

---

## [Author Response · Author response to Decision Letter 0]

31 Jan 2023

PONE-D-22-30476

Evaluation of Bread Wheat Genotypes for Drought Tolerance using Morpho-physiological Traits under Drought-stressed and Well-watered Conditions

PLOS ONE

Dear Dr. Sewore, 

Thank you for submitting your manuscript to PLOS ONE. After careful consideration, we feel that it has merit but does not fully meet PLOS ONE’s publication criteria as it currently stands. Therefore, we invite you to submit a revised version of the manuscript that addresses the points raised during the review process.

Especially consider the suggestions made by reviewers regarding statistical analysis which will improve the quality of the data being presented.

We look forward to receiving your revised manuscript.

Kind regards,

Rattan Singh Yadav, PhD

Academic Editor

PLOS ONE

Response of Authors: Thank you for the opportunity to revise and resubmit our paper. We hope we have now carefully and successfully addressed all the comments. 

Journal Requirements:

Response: After careful editing, we believe we have met all PLOS ONE’s style requirements. 

Response: We apologize for the grant information provided in the ‘Funding Information’ and ‘Financial Disclosure’ sections do not match. We now have properly provided in ‘Funding Information’ section. 

"The funders had no role in the design of the study; in the collection, analyses, or interpretation of data; in the writing of the manuscript, or in the decision to publish the results."

"Author name: Birhanu Mecha

Grant number: no

African Union funded stipend, and field and lab research costs.

Response: We provided the amended Funding Statement in the cover letter as mentioned above. 

Response: We uploaded the morpho-physiological data set to the Supporting Information files. 

Response: We provided the morpho-physiological data set to the Supporting Information files and we described the Data Availability statement in cover letter. 

6. Please amend either the abstract on the online submission form (via Edit Submission) or the abstract in the manuscript so that they are identical.

Response: We amended that the abstract on the online submission form and the abstract in the manuscript are identical. 

Response: We included one-line title captions for tables and figures as well as S1 Appendix”, “S2 Appendix so forth for supporting information files at the end of manuscript. 

Additional Editor Comments:

Your manuscript is interesting, however it needs revising. Please can you consider reviewers comments and resubmit for further considerations.

Reviewers' comments: 

Reviewer's Responses to Questions

Comments to the Author

1. Is the manuscript technically sounds, and do the data support the conclusions?

Reviewer #1: No

Reviewer #2: Partly

Response: We conducted the experiments using 196 genotypes (sample sizes) including 8 standard checks, lattice design with 2 replications. We concluded appropriately based on the data that was presented in the results. ________________________________________

2. Has the statistical analysis been performed appropriately and rigorously?

Reviewer #1: No

Reviewer #2: Yes

Response: Thank you dear reviewers for the kind feedback. The morphological data provided in supportive files were analysed separately for the drought-stressed and well-watered treatments computed using IBM SPSS (2022) Statistics version 28.0.1.1. We performed data on physiological traits separately and in combined for the drought-stressed and well-watered treatments using SAS and R (CRAN R Software package version 0.97, R Core Team, 2022). (See line 214-222). ________________________________________

3. Have the authors made all data underlying the findings in their manuscript fully available?

Reviewer #1: Yes

Reviewer #2: No

Response: We provided all the relevant data as part of the manuscript and its supporting information. ________________________________________

4. Is the manuscript presented in an intelligible fashion and written in Standard English?

PLOS ONE does not copyedit accepted manuscripts, so the language in submitted articles must be clear, correct, and unambiguous. Any typographical or grammatical errors should be corrected at revision, so please note any specific errors here 

Reviewer #1: Yes

Reviewer #2: No

Response: We corrected conclusion section of a paper to be brief and in accordance with the text and data discussed in the results and discussion section. ________________________________________

5. Review Comments to the Author

Response: We declared to the journal that the manuscript is not published elsewhere and not submitted to other journals. We declared there is no any conflict of interest.

Reviewer #1: The manuscript by Sewore et al. describes evaluation of 196 wheat genotypes under drought stress and optimum irrigation conditions in Ethiopia. The study is important and is of significant to Ethiopian breeders and to wheat scientists working on improving drought tolerance in wheat. However, there are many issues with the analysis and the presentation of results. Also, there are some fundamental errors, some of which can’t be improved just by revision.

1. Measurement of heritability’s of traits is missing.

2. Authors should add a figure of box plots of traits under optimum and drought stress conditions and use error bars.

3. Plant height and flowering time/days to heading are important traits to measure in drought stress trials. Is there any specific reason for not measuring these in the study?

4. BLUPs should have been calculated instead of just means in field experiments for all traits.

5. Authors should have created UPGMA trees to understand relationships among genotypes under both conditions separately and combined and should have discussed these relationships

6. If the objective is to characterize genotypes under drought and optimum irrigation conditions, why disease resistance scores were taken? What is the significance of adding disease resistance scores is not clear.

7. The Results and Discussion is poorly written and lacks synthesis

Response: Thank you so much for your constructive comments and review of our manuscript. This manuscript is one part of our study based on the drought tolerance potential of the tested genotypes focused on response of morpho-physiological traits to drought-stress and compare and select genotypes based on tolerance. We measured broad sense heritability of traits separately under drought-stressed and well-watered conditions. Agronomic traits such as Days to heading, plant height days to maturity, spike length, spikelets per spike, kernels number per spike, thousand kernels weight, and grain yield were well studied and prepared for publication in the next manuscript. We calculated BLUPs and created UPGMA trees for physiological traits (Please see supplementary information Figures 2 and 3). We added disease severity score to see the genotypes disease resistance performance under drought-stressed and well-watered conditions; drought-tolerant or drought-susceptible genotypes were resistance to disease. We revised the results and discussion part very well. We separated results from discussion and the results discussed in well organised way keeping synthesis. 

Reviewer #2: I have completed the review of this manuscript describing the evaluation of bread wheat genotypes under drought stress and well-watered conditions. Significant revisions should be made to the submitted manuscript before it could be considered acceptable for publication. Please address the specific points noted below in a revised manuscript.

Line no 226: what range used for size grouping give in details? Because all should not have the same size

Response: The tested genotypes flag leaf size was grouped based on its length and width category as described in the following table.

Flag leaf length Flag leaf width Leaf size (Morphology) Score

>15.25 cm >1.25 cm Very large 9

10.01 to 15.25 cm 1.01 to 1.25 cm Large 7

8.76 to 10.00 cm 0.76 to 1.00 cm Intermediate 5

<8.75 cm <0.75cm Small 3

Line no 228: It seems that 22 genotypes with very large leaf size categorized into 56 genotypes under stress condition? Give details of which genotypes leaf size reduced and level of change in leaf size in percentage. It will help to identify the better genotypes.

Response: There was no very large size flag leaf category under drought-stressed condition. Under drought-stressed condition, no genotype had very large flag leaf size. However, under well-watered condition, the four categories of leaf sizes were found: very large (22 genotypes), large (69 genotypes), intermediate (78 genotypes), and small (27 genotypes) (Please see: Line no 228 and Line no 229). The genotypes were indicated in supplementary table 1 (S1). The list of 22 genotypes with very large flag leaf size categorised under well-watered condition were: Tossa, Ogolcho, Dure, Pavon-76, Gambo, Enkoy, ETBW 8908, ETBW 8800, ETBW 9027, ETBW 9435, ETBW 8659, ETBW 9422, ETBW 8800, ETBW 9027, ETBW 9435, ETBW 9422, ETBW 8659, ETBW 9305, ETBW 9473, ETBW 9294, ETBW 9484, ETBW 9470, ETBW 8772, ETBW 9184, ETBW 8653, ETBW 8882 and LEMU. 

Line no 229: Author only gave graphical representation, give leaf pictures of also.

Response: We gave the leaf pictures of flag leaf size in supplementary information files. 

Line no 235-237: Part of flag leaf angle

Response: Thank you for your kind correction. It was shifted to flag leaf angle part. 

Line no 241: Again, give leaf figure showing different leaf angles. What leaf to stem angle used for three different categories?

Response: We gave the picture of flag leaf angle in supplementary information file. Flag leaf to stem angle was less than 90°, categorised into erect angle, between 90 and 180° categorised into semi erect and greater than 180° categorised in to droopy. 

Line no 242-244: According to the results under drought condition number of genotypes for both droopy and erect were almost same (51 and 48) but under well-watered condition genotypes with droopy leaf angel were around 2.5 times more than number of genotypes with erect leaf angle.

Line no 244: As per study drooping occurs due to two main reasons: water stress and over watering

In this study drooping genotypes number should be increased under stress condition relative to well water conditions. Results reveal that over watering was there.

Response: Thank you for your comments and correction. We included the comments in the manuscript. (See line 241-243) 

Line no 246-248: Early leaf dropping is a good example of drought avoidance mechanism, why author not considered it for drought resistance scoring.

Response: In our study, we observed the genotypes with erect leaf were more resistance to drought than droopy leaved. Thank you for guidance and we agree about future research in larger working groups.

Line no 250-251: Yes, true in case of yield but if you are handling with drought stress

Response: Thank you. 

Line no 253-255: Again, under stress condition number of genotypes with leaf rolling is least, however, increased under well-watered condition. Authors give data in a random order. Always represent data in ascending or descending order. Write manuscript in a scientific way, under stress condition start results with data for no flag leaf rolling however, under well-watered condition started with flag leaf rolling.

Response: Sorry for the missing of “no” in manuscript. We corrected it as: Genotypes under well-watered condition showed that 72 (36.7%) of genotypes did exhibit no flag leaf rolling, 64 (32.7%) of genotypes had weak rolling, 32(16.3%) of genotypes had semi-rolling, whereas 28 (14.3%) genotypes manifested full rolling (Figure 2C).

PCA: Discuss grouping of genotypes on the basis of their pedigree. Genotypes within the group have same pedigree?

Response: No. They have different pedigree. But we listed them as we obtained from the Ethiopian institute of Agricultural research. 

Fig 1 and 2; number in graphs overlapping, should use small font size.

Response: Corrected 

Appendix C: give details of genotypes, their pedigree etc.

Response: we listed them as provided from the Ethiopian institute of Agricultural research (EIAR). We gave pedigrees for some of them. ________________________________________

6. PLOS authors have the option to publish the peer review history of their article (what does this mean?). If published, this will include your full peer review and any attached files.

Do you want your identity to be public for this peer review? For information about this choice, including consent withdrawal, please see our Privacy Policy.

Reviewer #1: No

Reviewer #2: No

Response: Dear Reviewers, thank you for the kind feedback. Related to your comments and corrections we elaborated on our findings. Thank you!

---

## [Editor Report · Decision Letter 1]

7 Mar 2023

Evaluation of Bread Wheat <triticum aestivum="" l.=""> Genotypes for Drought Tolerance using Morpho-physiological Traits under Drought-stressed and Well-watered Conditions

PONE-D-22-30476R1</triticum>

Dear Dr. Sewore,

Many thnaks for revising your ms based on reviwers comments.  We’re pleased to inform you that your manuscript has been judged scientifically suitable for publication and will be formally accepted for publication once it meets all outstanding technical requirements.

Kind regards,

Rattan Singh Yadav, PhD

Academic Editor

PLOS ONE
---

## [Editor Report · Acceptance letter]

24 Apr 2023

PONE-D-22-30476R1 

Evaluation of Bread Wheat (*Triticum aestivum* L.) Genotypes for Drought Tolerance using Morpho-physiological Traits under Drought-stressed and Well-watered Conditions 

Dear Dr. Sewore:

I'm pleased to inform you that your manuscript has been deemed suitable for publication in PLOS ONE. Congratulations! Your manuscript is now with our production department. 

Kind regards, 

on behalf of

Dr. Rattan Singh Yadav 

Academic Editor

PLOS ONE